# Key activity descriptors of nickel-iron oxygen evolution electrocatalysts in the presence of alkali metal cations

Mikaela Görlin [1,2✉], Joakim Halldin Stenlid [1], Sergey Koroidov[1], Hsin-Yi Wang [1], Mia Börner[1], Mikhail Shipilin[1], Aleksandr Kalinko [3,4], Vadim Murzin [4,5], Olga V. Safonova [6], Maarten Nachtegaal [6], Abdusalam Uheida[7], Joydeep Dutta [7], Matthias Bauer[3], Anders Nilsson[1] & Oscar Diaz-Morales [1,8✉]

Efficient oxygen evolution reaction (OER) electrocatalysts are pivotal for sustainable fuel production, where the Ni-Fe oxyhydroxide (OOH) is among the most active catalysts for alkaline OER. Electrolyte alkali metal cations have been shown to modify the activity and reaction intermediates, however, the exact mechanism is at question due to unexplained deviations from the cation size trend. Our X-ray absorption spectroelectrochemical results show that bigger cations shift the $Ni^{2+/(3+\delta)+}$ redox peak and OER activity to lower potentials (however, with typical discrepancies), following the order $CsOH > NaOH \approx KOH > RbOH > LiOH$. Here, we find that the OER activity follows the variations in electrolyte pH rather than a specific cation, which accounts for differences both in basicity of the alkali hydroxides and other contributing anomalies. Our density functional theory-derived reactivity descriptors confirm that cations impose negligible effect on the Lewis acidity of Ni, Fe, and O lattice sites, thus strengthening the conclusions of an indirect pH effect.

[1] Department of Physics, AlbaNova University Center, Stockholm University, SE-106 91 Stockholm, Sweden. [2] Department of Chemistry - Ångström laboratory, Uppsala University, Box 538, SE-751 21 Uppsala, Sweden. [3] Department of Chemistry and Center for Sustainable Systems Design (CSSD), University of Paderborn, Warburger Strasse 100, D-33098 Paderborn, Germany. [4] Deutsches Elektronen-Synchrotron DESY, Notkestraße 85, D-22607 Hamburg, Germany. [5] Bergische Universität Wuppertal, Gaußstraße 20, D-42119 Wuppertal, Germany. [6] Paul Scherrer Institute, CH-5232 Villigen, Switzerland. [7] Functional Materials, Department of Applied Physics, School of Engineering Sciences, KTH Royal Institute of Technology, Hannes Alfvéns väg 12, SE-114 19 Stockholm, Sweden. [8] Applied Electrochemistry, School of Chemical Science and Engineering, KTH Royal Institute of Technology, SE-100 44 Stockholm, Sweden. ✉email: mikaela.gorlin@kemi.uu.se; oadm@kth.se

To slow down the growth of the steadily increasing carbon footprint, a transition to renewable energy is imperative[1]. Electrochemical water splitting ($2\,H_2O \rightarrow 2\,H_2 + O_2$) offers a zero-carbon route to hydrogen ($H_2$) from water[2], where the main cause of energy loss and cost in this process is the anodic oxygen evolution reaction (OER)[3]. To make water a viable source for $H_2$, efforts therefore need to be focused on more efficient OER electrocatalysts[4].

Here, we investigate the OER activity and redox-activity using X-ray absorption spectroscopy (XAS) of a Ni–Fe oxyhydroxide (OOH) electrocatalyst in the presence of alkali metal cations ($Li^+$, $Na^+$, $K^+$, $Rb^+$, $Cs^+$), which is one of the best performing catalysts in alkaline media[5,6]. Electrolyte cations are known to impact the oxygen evolution activity of various oxide-derived electrocatalysts[7–12], where the activity seemingly follows the trend in cation size, typically increasing from small ($Li^+$) to large ($Cs^+$) cations. However, the role of cations in OER is not entirely understood due to reoccurring discrepancies from the trend in cation size. Moving down the alkali metal group, the cation size increases from $Li^+$ to $Cs^+$, along with modifications of several parameters such as decreasing Lewis acidity and electronegativity[13], increasing molar conductivity[14] and proton affinity[15], and increasing basicity of the corresponding alkali hydroxide[16]. Small cations ($Li^+$) generally form stronger non-covalent interactions with water compared to large cations ($Cs^+$), and water becomes more structured around the central cation[17]. Smaller cations are therefore referred to as "structure makers" and bigger cations as "structure breakers". The strong non-covalent interactions between small cations and water disrupt the native H-bonded network and result in large solvation shells[17], which is usually associated with slower reorientation times and slower kinetics[14], and observed to alter redox-kinetics of metal-centers[18]. However, there are several unexpected deviations from the cation size trend. Michael et al.[19] showed that the activity of the NiOOH catalyst increased from LiOH to CsOH, whereas the activity of Ni(Fe)OOH was lower in CsOH compared to both NaOH and KOH. Zaffran et al.[7] demonstrated using density functional theory (DFT) that electrolyte cations modify the adsorption energies of OER intermediates (*OH, *O, *OOH) of the Ni–Fe catalyst, where especially small and strongly acidic cations are not beneficial for OER. Garcia et al.[8] found using surface enhanced Raman spectroscopy that large cations promote peroxo-like "active oxygen" species ($O^-$ or $O_2^-$) in NiOOH to a larger extent than small cations, which could explain the higher OER activity in the presence of large cations. Yet, several inconsistencies in the activity trends put the mechanism by which the alkali metal cations modify the OER activity at question.

The catalytic site in the Ni–Fe catalysts has been characterized using in situ XAS, where several studies reveal contradicting information regarding the impact of Fe on the redox-activity of the Ni-site[20–27]. According to several DFT studies, the Fe-site plays a significant role as a low overpotential-site in the bimetallic Ni–Fe active site and provides optimal adsorption energies for the OER intermediates[28–31], which is also supported by experiments[32]. In addition, Fe at coordinatively unsaturated sites such as edge sites or defect sites are predicted as more reactive[33–35]. Recent studies employing in situ soft XAS at the O $K$-edge also confirmed anionic redox-activity involving the lattice oxygens in the Ni–Fe catalyst[36,37], most likely related to the "active oxygen" earlier identified in Raman spectroscopy[38–41].

Here, we employ in situ XAS at the Ni and Fe $K$-edges to probe the local atomic structure and metal redox-states of an electrodeposited $Ni_{65}Fe_{35}$(OOH) catalyst in the presence of alkali metal cations at alkaline pH (i.e. LiOH, NaOH, KOH, RbOH, CsOH). We further utilize DFT to explore the correlations between the OER activity and three reactivity properties: The local electron

attachment energy $E(\mathbf{r})$[42], the local average ionization energy $\bar{I}(\mathbf{r})$[43], and the electrostatic potential $V(\mathbf{r})$, to predict how electrolyte cations influence the local Lewis acidity/basicity of the Ni–Fe(OOH) lattice sites[44–46]. In short, our data conclude that the modification of alkali metal cations on the OER activity can be explained as a response to a change in the electrolyte pH.

## Results

### The impact on the OER activity of the Ni-Fe catalyst by alkali metal cations.

A demonstration of the cell setup used for electrochemical and in situ XAS investigations of the electrodeposited $Ni_{65}Fe_{35}$(OOH) catalyst is shown in Fig. 1a, b. The OER activity of the $Ni_{65}Fe_{35}$ catalyst was evaluated in 0.1 M alkali hydroxide solutions (LiOH, NaOH, KOH, RbOH, CsOH), purified according to a modified protocol by Boettcher and co-workers (Supplementary Note 1)[47]. Scanning electron microscopy (SEM) confirmed a platelet-like layered morphology of the $Ni_{65}Fe_{35}$ catalyst (Fig. 1c)[47]. The metal loading was estimated to ~25 µg cm$^{-2}$, corresponding to a film thickness of ~300 nm. Cyclic voltammograms (CVs) and activity trends from steady-state data during the in situ XAS measurements are presented in Fig. 1d, e. (The spectra will be presented in the XAS section below.) We find that the activity increases in the order of LiOH < RbOH < KOH < NaOH < CsOH, thus does not exactly follow the overall size of the cations (see Fig. 1e). A larger data set recorded "in house" largely confirms the activity trend from the XAS data set, where the OER overpotential ($\eta_{OER}$) at 10 mA cm$^{-2}$ increases in the order of CsOH (295 mV) < NaOH (309 mV) < KOH (316 mV) < RbOH (331 mV) < LiOH (352 mV), see Fig. 2a[6].

Since the two data sets (XAS vs. "in house") were measured using two different reference electrodes (Ag/AgCl vs. RHE), the trends are validated in Supplementary Fig. 1. The Tafel slopes are in line with the overall activity trends, decreasing from 41 mV dec$^{-1}$ in LiOH to 38 mV dec$^{-1}$ in CsOH, and with the typical discrepancies in between the two extremes (see Fig. 2b and Supplementary Fig. 1f). The turnover frequency was estimated based on the total metal loading (TOF$_{mass}$) including both the amount of Ni and Fe determined by elemental analysis (ICP-OES), see the Methods section for more details. The TOF$_{mass}$ is in good accord with the geometric activity trend, and increases in the order of LiOH (0.009 s$^{-1}$), RbOH (0.014 s$^{-1}$), KOH (0.015 s$^{-1}$), NaOH (0.016 s$^{-1}$), and CsOH (0.023 s$^{-1}$) at 1.56 V in purified 0.1 M alkali hydroxides (more data is shown in Supplementary Fig. 1 and activity parameters in Supplementary Table 1). The TOF$_{mass}$ appears fairly low at first glance, however, agrees well with reports of other Ni–Fe catalysts at similar conditions[6], where the turnover frequency is generally lower for thicker films and in lower electrolyte concentrations, as we also show below[48,49]. Although the cation trends herein agree with the reported trends for Ni–Fe catalysts, there are several discrepancies between these studies[7,8,19,40]. As mentioned Michael et al.[19] found highest activity of their NiOOH catalyst in CsOH in Fe-free conditions, however, when Fe-impurities were added to the electrolyte, the activity in CsOH decreased drastically. On the contrary to most of these studies, Ding et al.[9] found that the activity was highest in LiOH for a series of transition metal oxides (Co, Fe, Mn), which was regarded to a suppression of the activity for the oxygen reduction reaction (ORR), which is the "backward" reaction for OER. Suntivich et al.[50] also found a generally lower ORR activity in LiOH, however, the OER activity was still lowest in LiOH in line with other studies.

The activity drop observed in RbOH we cannot find support for due to the lack of reports for Ni–Fe catalysts. On the contrary, a higher than expected activity was found in RbOH on Pt(111) in a study by Tymoczko et al.[11], although the explanation for this

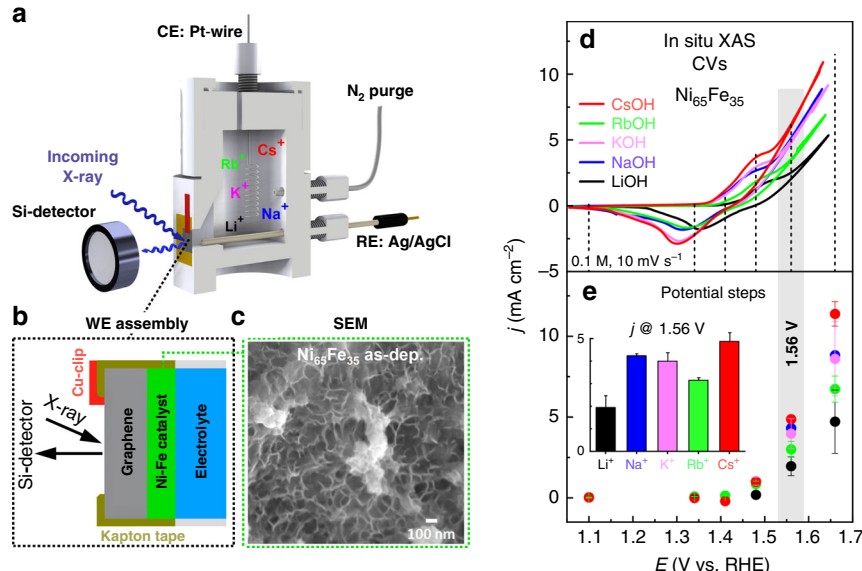

**Fig. 1 Electrochemical cell setup and OER activity during in situ XAS. a** Schematic drawing of the cell used for the in situ XAS investigations of the $Ni_{65}Fe_{35}(OOH)$ catalyst, carried out in purified 0.1 M LiOH, NaOH, KOH, RbOH, and CsOH. The cell body was made of PTFE/PEEK ($\varnothing = 50$ mm, with a 6 × 6 mm X-ray transparent window), a leak-free Ag/AgCl reference electrode, and a Pt-mesh as counter electrode. **b** Enlargement of the working electrode showing the ~300 nm thick $Ni_{65}Fe_{35}$ film electrodeposited on a ~35 μm thick graphene-sheet as working electrode. **c** SEM image of the as-deposited $Ni_{65}Fe_{35}$ film on the graphene electrode. **d** CVs at 10 mV s$^{-1}$ during the in situ XAS measurements at the P64 beamline at Petra III (DESY). **e** Potential steps during collection of the Ni and Fe K-edges, where the inset shows a trend plot of current density ($j$) at 1.56 V against the electrolyte cation. All potentials have been converted from the Ag/AgCl to the RHE potential scale. Error bars represent the standard error from the overall number of measurements.

remained ambiguous. Here we observe that electrolyte cations have a clear effect on the potential of the nickel redox peak, located around 1.5/1.3 V vs. RHE in the CVs (Fig. 2c). This is assigned to the reversible oxidation/reduction of $Ni^{2+} \leftrightarrow Ni^{3+/4+}$ (which we denote "$Ni^{2+/(3+\delta)+}$" since the fraction of $Ni^{3+/4+}$ depends on the potential) via deprotonation of lattice hydroxide (–OH) to oxidic groups (–O)[38]. The anodic redox peak ($E_{p,a}$) shifts to higher potentials by ~40 mV from CsOH to LiOH, which agrees with the shift of the OER activity in the same direction (see Supplementary Fig. 1g–i). The integrated anodic peak charge ($Q_{p,a}$) is only ~0.5 e$^-$ per Ni, whereas the cathodic peak charge ($Q_{p,c}$) is closer to ~1 e$^-$ per Ni since it is well-separated from the catalytic current (see Fig. 2c and Supplementary Fig. 1g–i)[51]. A similar cation effect has been recognized by Yang et al.[40] where the tetrabutylammonium (TBA$^+$) and tetramethylammonium (TMA$^+$) cations (thought to interact with "active oxygen") shifted the redox peak similarly by ~78 mV to higher potentials. Such modifications were also recognized by Strmcnik et al.[10] for metallic Pt, where strongly adsorbing Li$^+$ ions were proposed to form clusters with surface adsorbed *OH species, and thereby blocking the active sites. Suntivich et al.[50] instead proposed that Li$^+$ ions interfere with the rate-limiting steps/species in the catalytic cycle via strong non-covalent interactions, and thereby impact the reaction energy and the kinetics. This is in line with the DFT study by Zaffran et al.[7], where electrolyte cations (alkali and alkali earth) were found to influence the adsorption energies of the OER intermediates.

**Catalyst-cation interactions and electrolyte impurities**. SEM analysis post-OER did not indicate any significant change in the morphology of the Ni–Fe catalyst (Supplementary Fig. 2). Energy-dispersive X-ray spectroscopy (EDS) further supports a non-covalent catalyst-cation interaction, since there were no traces of cations in the Ni–Fe films post-OER after carefully rinsing with MilliQ-water (Supplementary Fig. 3a, b). Only in

films not rinsed as carefully, traces of alkali cations were occasionally detected, which is more likely traces of dried hydroxide salts. The Ni–Fe catalyst films on the other hand were found to exhibit a significant local inhomogeneity in the Ni:Fe composition, where several local areas have a clear Fe-enrichment (Supplementary Fig. 3c, d and Supplementary Table 2). Such local compositional inhomogeneity has earlier been reported beyond ~25% Fe-content for Ni–Fe catalysts[20,52]. Since these variations were also present in the as-deposited Ni–Fe films, it at least excludes that this is related to a specific alkali cation. Furthermore, electrolyte impurities (such as Fe) might incorporate to surface sites and thereby affect the catalytic activity and redox-peak system of Ni-based catalysts, whereby we used ICP-OES (and EDS) to scan the alkali hydroxides. Impurities were found only in the as-received electrolytes; ~5 ppm Fe and ~3 ppm Zn in RbOH, and ≤1 ppm Fe in CsOH (Supplementary Fig. 4a), whereas no impurities were found above the detction limit in the purified hydroxides, suggesting that the purification step was successful (Supplementary Fig. 4b, c).

**In situ XAS in presence of alkali metal cations**. In situ XAS was acquired at the Ni and Fe K-edges in purified 0.1 M alkali hydroxides at the P64 beamline at Petra III (DESY, Hamburg). The Ni or Fe K-edge XAS probes dipole-allowed transitions from the 1 s core level to unoccupied 4p valence states of the 3d metals, and the pre-edge feature ~10 eV below the edge is assigned to quadrupole transitions to empty 3d states[53]. In Fig. 3a, b, the X-ray absorption near edge spectra (XANES) of the ~300 nm thick $Ni_{65}Fe_{35}$ catalyst film are shown in the non-catalytic "ground" state (1.1 V), and at a selected potential in the half-oxidized state after the redox-onset (1.48 V) where the differences between the metal cations were largest (see also Supplementary Figs. 5–8). At non-catalytic potential, and prior to the onset of metal oxidation (i.e., between 1.1 V–1.34 V vs. RHE), the K-edge positions match low-valent $Ni^{2+}Fe^{3+}$, independent of the electrolyte cation

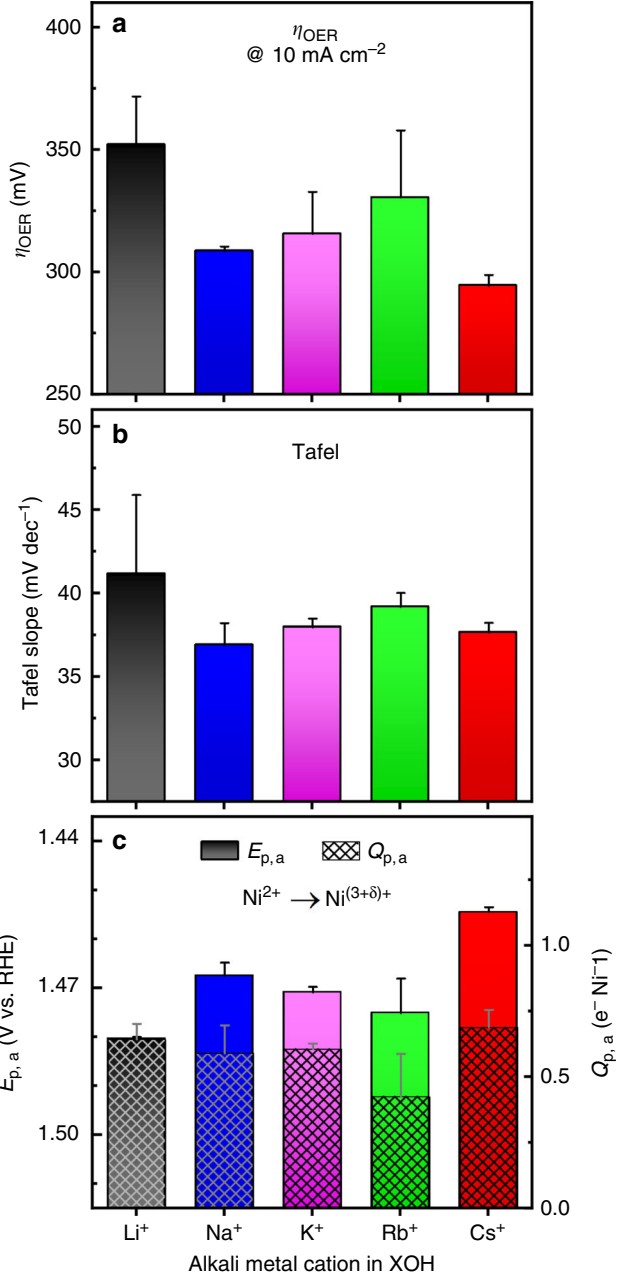

**Fig. 2 OER activity trends at the influence of alkali metal cations.** The OER activity of the $Ni_{65}Fe_{35}$ catalyst in purified 0.1 M XOH electrolytes (X = Li$^+$, Na$^+$, K$^+$, Rb$^+$, and Cs$^+$) measured "in house". **a** OER overpotential ($\eta_{OER}$) at 10 mA cm$^{-2}$ from steady-state conditions. **b** Tafel slope analysis extracted from steady-state conditions. **c** The anodic peak positions ($E_{p,a}$) of the Ni$^{2+}$→ Ni$^{(3+\delta)+}$ redox peak (left), and the corresponding integrated anodic peak charge ($Q_{p,a}$, right) obtained from CVs at 10 mV s$^{-1}$. All data in this figure was recorded using a reversible hydrogen electrode (RHE) as a reference electrode. Error bars show the standard error from the overall number of measurements.

(see Fig. 3e; Supplementary Fig. 9a, b and Supplementary Table 3). Simulations of the $k^3$-weighted extended X-ray absorption fine structure (EXAFS) using scattering functions generated in FEFF[54] show that Ni–O is coordinated at a distance of ~2.04 Å, Ni–Ni at ~3.06 Å, Fe–O at ~1.99 Å, and Fe–Ni at ~3.06 Å, which match the oxidation states from the edge positions, consistent with a Ni$^{2+}$Fe$^{3+}$ ground state (see Fig. 3c, d; Supplementary Fig. 9c, d; Supplementary Tables 4 and 5). Hence, no distinct differences are observed for different electrolyte

cations in the non-catalytic state. Above the oxidation threshold (above ~1.35 V), the Ni K-edge starts to shift to higher energies, consistent with an increase in the metal oxidation state from Ni$^{2+}$ to a half-oxidized state between Ni$^{3+}$ and Ni$^{4+}$ (i.e., "Ni$^{(3+\delta)+}$"), however, the absolute edge shift depends on the electrolyte cation (see OX region in Fig. 3e). At 1.48 V, the differences between the cations are largest, and the Ni K-edge positions increase in the order LiOH < RbOH < KOH ≈ CsOH < NaOH (Supplementary Fig. 10a–c). This is similar to the characteristic trend seen in the electrochemical characterization, except that NaOH has a larger than expected edge shift, which we will come back to later. Conversely, the Fe K-edge does not shift at the main edge, and all Fe-sites therefore appear to remain as Fe$^{3+}$ independent of the potential and electrolyte cation. Although, there is a small increase in the pre-edge intensity and a positive shift of the white-line centroid as observed in previous studies (Supplementary Fig. 6)[20,21,23,25]. At higher potentials with a stronger catalytic rate (1.66 V), the Ni K-edge is consistent with an overall oxidation state of Ni$^{3.8+}$, meaning that all sites have oxidized to Ni$^{3+}$, and further ~80% of these to Ni$^{4+}$, where the differences between the cations are again smaller (Fig. 3e and Supplementary Table 3). This is in large agreement with previous work[20], however, the oxidation states of both sites (Ni and Fe) are disputed and varies across studies[21–23,25,48]. The simulations of the $k^3$-weighted EXAFS confirms a shortening of both the Ni–O and Fe–O bonds at oxidative potential, and at 1.66 V the following average bond lengths are obtained; Ni–O (~1.88 Å), Ni–Fe (~2.82 Å), Fe–O (~1.94 Å), and Fe–Ni (~2.89 Å), although, the absolute bond lengths depend on the electrolyte cation (Fig. 3f and Supplementary Tables 4 and 5). The Ni–O distances are consistent with an oxidation state of Ni$^{3.7+}$, thus largely agree with the Ni K-edge positions (Supplementary Figs. 9c and 10). The short Fe–O bond on the other hand contradicts the Fe K-edge position, and is consistent with ~60–90% of the Fe-centers being oxidized to Fe$^{4+}$ at 1.66 V, depending on the cation (Supplementary Figs. 9d and 10 and Supplementary Table 5). Michael et al.[19] also observed in Raman spectroscopy that CsOH promotes a longer Ni–O bond in the Ni(Fe)OOH catalyst compared to LiOH. Our XAS data herein instead supports a shorter Ni–O bond, which just reflects the relatively higher oxidation state of Ni in CsOH. Although there are some differences between the cations, the correlation between the Fe–O bond length, electrolyte cation, and OER activity is not particularly strong. This does not necessarily rule out Fe as the active site, but might suggest that the Fe-site is not rate-limiting for the turnover rate in contrast to the Ni-site. The changes related to the Fe XAS spectra are in fact a matter of controversy. Dau and co-workers did not regard the spectral change to an increase in the Fe oxidation state due to the lack of an edge shift, but instead proposed geometric distortions[21,23]. Although, herein we observe a strong contraction of the Fe-O bond, which is usually associated with an increase in the oxidation state. Other spectroscopic techniques also support a fraction (3–21%) of high-valent Fe-species (+4 and beyond) being formed during OER[26,55]. Another possible explanation, as mentioned by Hunter et al.[56], is a spin-state transition from high-spin Fe$^{3+}$ to low-spin Fe$^{3+}$ [57], which could result in a similar spectral change as both geometry and charge-transfer effects, the latter proposed for Ni-Fe catalysts[25]. This will have to be addressed in future studies.

Noteworthy, the differences between the cations are most pronounced at quasi-equilibrium conditions (in the OX region in Fig. 3e) and diminish at higher potentials (in the OER region). This suggests that the electrolyte cations do not significantly alter the equilibrium levels of oxidized Ni$^{(3+\delta)+}$, but rather shifts the onset potential of oxidation. Repetitions and a cross-check with another data set from a different beamline (SuperXAS-XD10 at

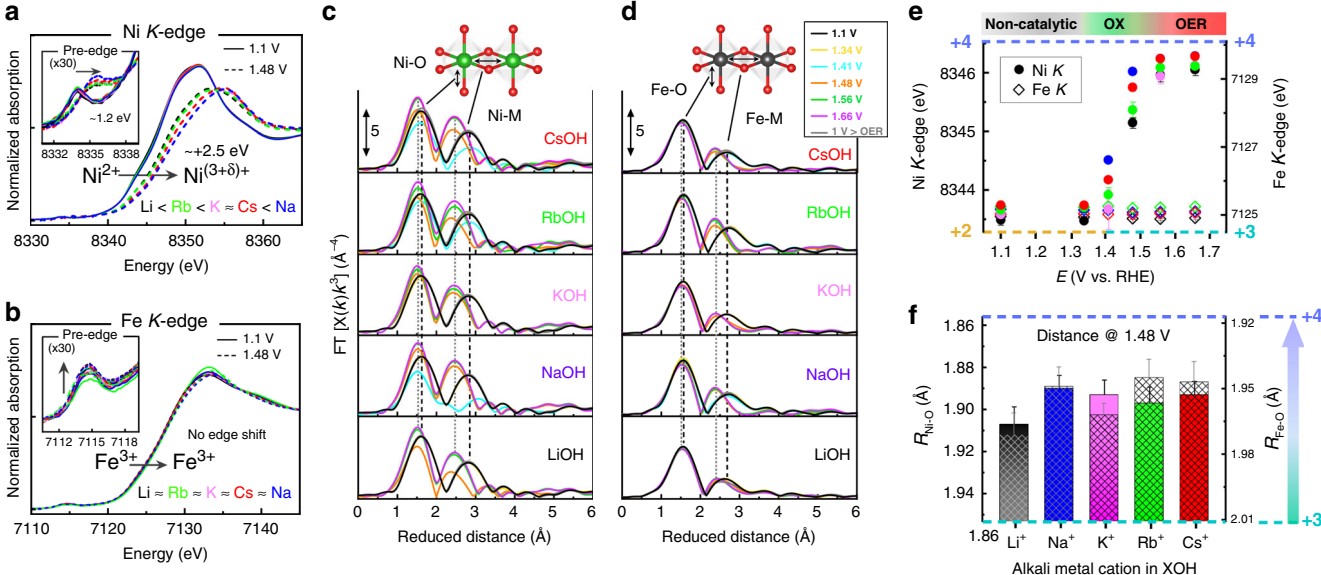

**Fig. 3 In situ X-ray absorption spectroscopy in presence of alkali metal cations.** In situ XAS was performed at the Ni and Fe K-edges of the $Ni_{65}Fe_{35}(OOH)$ catalyst in purified 0.1 M XOH electrolytes (X = $Li^+$, $Na^+$, $K^+$, $Rb^+$, $Cs^+$). **a** Ni XANES at 1.1 V (solid curves) and at 1.48 V (dashed curves). The inset shows an enlargement of the pre-edge region and the arrow indicates the direction of change upon oxidizing potential. **b** Fe XANES at 1.1 V (solid curves) and at 1.48 V (dashed curves). The inset shows an enlargement of the pre-edge region with the direction of change indicated by the arrow. **c** The $k^3$-weighted FT-EXAFS at the Ni K-edges. **d** The $k^3$-weighted FT-EXAFS at the Fe K-edges. **e** Ni and Fe K-edge positions vs. the applied potential. The colored areas on top indicate the different regions (non-catalytic, oxidative "OX", and "OER" catalytic). **f** Trend-plots of the Ni–O and Fe–O coordination distances at 1.48 V vs. the alkali metal cation (X) in the XOH electrolyte. Oxidation states are indicated in **e** and **f** with following colors: +2 (yellow), +3 (green), and +4 (purple). Error bars show the standard error from the overall number of measurements.

SLS, PSI) confirms that the cation-induced differences in the Ni K-edge positions are relatively small, and in fact close to the overall standard error (Supplementary Fig. 11). Overall, the second data set supports that the fraction of oxidized $Ni^{(3+\delta)+}$ increases from small to large cations, where the lowest and highest oxidation states are found in LiOH and CsOH, respectively. However, we find that the absolute edge positions are influenced by the catalyst film thickness which is related to a shift in the OER overpotential with loading, which might therefore introduce further discrepancies in the edge positions between the cations. More information on the XAS data analysis is provided in Supplementary Note 2, and EXAFS simulations and calibration curves are shown in Supplementary Figs. 12–14.

**The electrolyte pH as an OER activity descriptor**. To address a possible pH-effect imposed by the alkali cations, we extended our investigations to a wider range of molar concentrations, namely 0.05 M, 0.1 M, 0.25 M, 0.5 M, and 1 M alkali hydroxides. A reversible hydrogen electrode (RHE) was employed as a reference electrode for these measurements to avoid unknown shifts in the potential-scale due to small variations in the pH[58]. Our results show that the $Ni_{65}Fe_{35}$ catalyst responds to an increased electrolyte concentration (and thus pH) with a shift of both the $Ni^{2+/(3+\delta)+}$ redox peak and the OER activity to lower potentials; this effect is remarkably similar to the effect seen from small to large cations (see Fig. 4a, b and Supplementary Fig. 15a). Both the peak positions ($E_{p,a}$) and the activity are found to shift in the same direction and with fairly similar magnitudes (~40–50 mV), and therefore the separation between these two processes on a potential-scale remains fairly constant (~70–95 mV). In Fig. 4c, it is evident that the activity increases significant with the molar concentration, however, smaller cations show a less steep increase with the concentration compared to large cations (see also Supplementary Fig. 15b, c). The differences therefore become more

apparent at higher concentrations. We also note that the persistent dip seen in 0.1 M RbOH is less pronounced in 1 M RbOH, hence showing that the absolute cation trend depends on the molar concentration (Supplementary Fig. 15d, e). The electrolyte pH was determined using a pH-meter, which confirms that the differences between the alkali metal cations indeed are also reflected in the pH and not only in the OER activity (Supplementary Fig. 15f and Supplementary Table 6). Therefore, a plot of the OER activity against the electrolyte pH, not surprisingly, almost entirely diminish the characteristic cation trend, where the activity instead increases from low to high electrolyte pH (see Fig. 4d, e), also confirmed in the intrinsic activity parameter ($TOF_{mass}$) at several electrode potentials (Supplementary Fig. 15h–m). Note that the pH in LiOH could not be determined with a pH-meter due to the small size of the $Li^+$ ion, which appears to have too similar transport properties through the membrane of the pH electrode as the proton. An approximate value of the pH in LiOH was instead obtained from the potential offset between the RHE and the Ag/AgCl reference electrodes (Supplementary Note 3 and Supplementary Fig. 16a, b). The activity in LiOH sometimes falls a bit off the linear trend, however, we regard this to the alternative method used for determining the electrolyte pH.

The fact that the pH appears to be a strong OER activity descriptor implies that the differences between the electrolyte cations can simply be explained as a response to a change in the electrolyte pH. These findings strongly suggest that what at first glance appears as a "specific" cation effect is more likely an indirect effect due to changes in the electrolyte pH, which has to the best of our knowledge not been recognized in earlier OER studies. An effect on the electrolyte pH by alkali cations has although been reported for the $CO_2$ reduction reaction (CO2RR) on metallic Ag and Au, where Sing et al.[59] showed that larger cations such as $Cs^+$ improve the buffer capacity, which prevents an increase in the local pH. This in turn decreases the faradaic

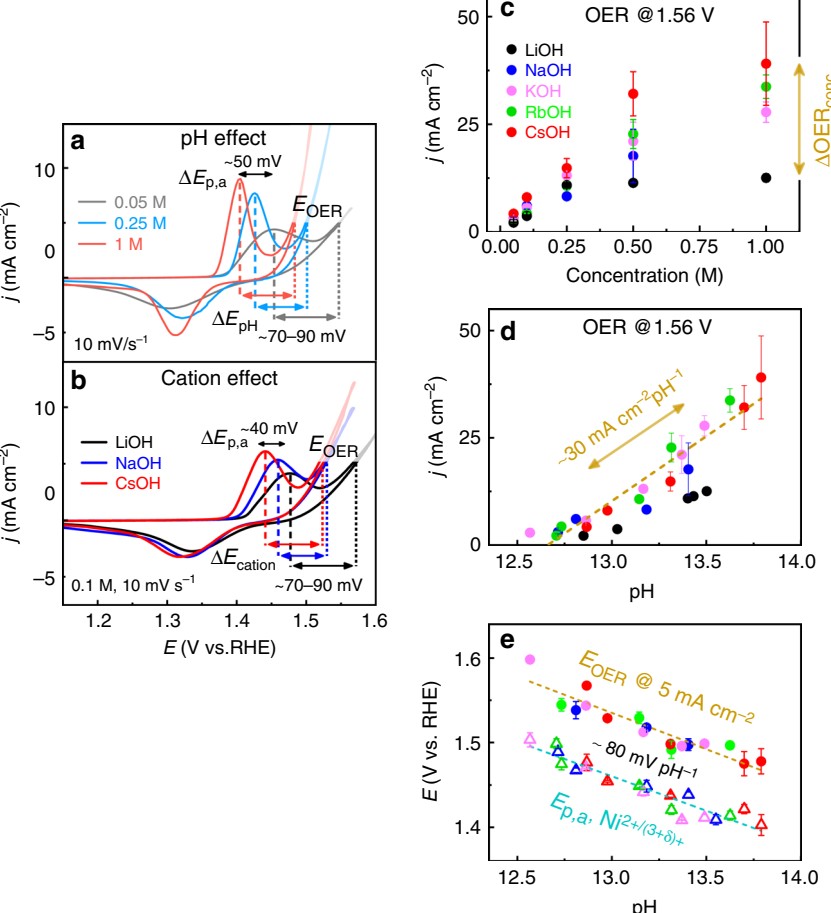

**Fig. 4 The electrolyte pH as OER activity descriptor. a** CVs at 10 mV s$^{-1}$ of Ni$_{65}$Fe$_{35}$(OOH) in 0.05 M, 0.25 M, and 1 M CsOH, to demonstrate the effect on the OER activity of increasing pH from low to high concentrations. **b** CVs at 10 mV s$^{-1}$ in 0.1 M LiOH, 0.1 M NaOH, and 0.1 M CsOH, to demonstrate the typical peak shift induced from small to large cations. **c** Current density ($j$) at 1.56 V vs. RHE plotted against the electrolyte concentration. **d** Current density at 1.56 V vs. RHE plotted against the electrolyte pH. **e** The OER potential ($E_{OER}$) at 5 mA cm$^{-2}$ (filled circles) and the anodic peak potential ($E_{p,a}$) of the Ni$^{2+}$ ↔ Ni$^{(3+\delta)+}$ redox peak from CVs at 10 mV s$^{-1}$ (hollow up-pointing triangles). The color code for the alkali metal hydroxides shown in the legend in **c** also applies to **d**, **e**. The pH in LiOH was determined with an alternative method (Supplementary Note 3), and is therefore excluded from the fit line in **d** and not shown in **e** for clarity. Error bars show the standard error from the overall number of measurements.

efficiencies (FE) toward H$_2$ and CH$_4$, and increases the FE for CO, C$_2$H$_4$, and C$_2$H$_5$OH. The pH effect for CO2RR cannot be directly translated to alkaline OER, however, we stress that the pH is a likely critical parameter for reactions involving proton-coupled electron-transfer steps[38].

An increase in the pH from LiOH to CsOH is in fact expected from their respective p$K_b$ values (Supplementary Table 6), following the increase in the basicity of the alkali hydroxide (and the decrease in Lewis acidity of the cation) from small to large cations[16,59]. We use the Henderson-Hasselbalch formalism to retrieve experimental pH values (Supplementary Note 4 and Supplementary Equations (1)–(13)). This indeed verifies that the pH is higher for more basic hydroxides (i.e., for bigger cations) due to a larger amount of dissociated OH$^-$ (Supplementary Fig. 15g and Supplementary Table 6)[16]. This also confirms that larger differences are to expect first at higher concentrations, while it therefore might be challenging to capture this effect unless a wider pH window is considered. Despite that theory predicts an increase in pH from small to large cations, it cannot directly explain the deviations from the cation size trend in, e.g., RbOH. To address this, we used ICP-OES to determine the cation concentrations in 0.1 M and 1 M electrolytes. This indeed shows a slightly lower than expected concentration of cations in both

RbOH and LiOH, thus reflecting the variations in activity (Supplementary Table 7). This points toward actual mistakes in the electrolyte concentration rather than to an increased amount of e.g. dissolved HCO$_3^-$, which could also lower the pH[7,60]. Since the salts are very hygroscopic (especially RbOH and CsOH), this could be one factor that contributes to further error in the concentration, as well as other dilution mistakes during preparation. It should be remembered that alkali metal cations are easily ionized elements (EIE) (especially Li$^+$), which can also cause uncertainties in ICP-OES analysis (see further details in the Methods section). To further verify the pH effect, we therefore carried out experiments where the pH had been adjusted to near-identical values for all alkali hydroxides. The results show that the OER activities indeed align more closely (or show a different trend) after adjusting the pH values (Supplementary Fig. 17). This concludes that the "characteristic" cation trend is indeed easily displaced by rather small changes in the electrolyte pH, which further supports a pH effect rather than a specific cation effect. All the OER activity parameters at different pH are listed in Supplementary Table 8.

We observe that both the Ni$^{2+/(3+\delta)+}$ redox-peak and the OER overpotential exhibit non-Nernstian pH-slopes of ca. −80 mV pH$^{-1}$ in the RHE scale (Fig. 4e). The absolute pH-slopes are although

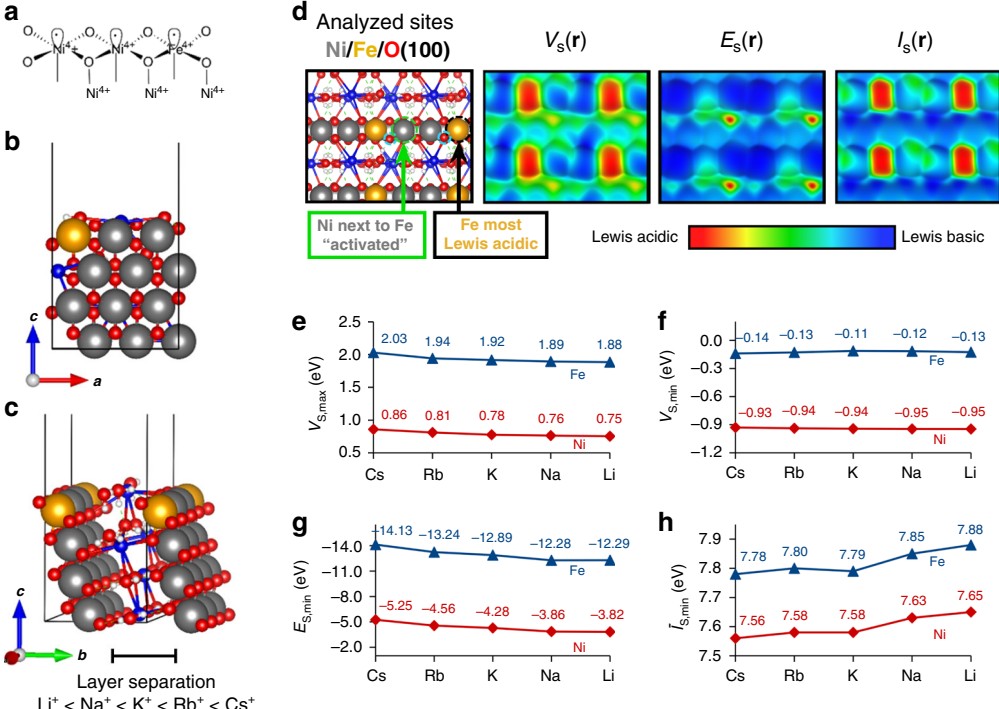

**Fig. 5 Computational analysis of the reactive site. a–c** Surface model of γ-Ni(Fe)OOH(100) with intercalating water and alkali metal cations. **d** Top-view of the surface with reactivity properties estimated using DFT mapped on the 0.001 au isodensity contour is shown in. **e–h** Local extrema in the properties of the most activated Ni/O and Fe/O sites, which indicate the Lewis acidity of Ni and Fe sites ($V_{S,max}$ and $E_{S,min}$) and Lewis basicity of O sites ($V_{S,min}$ and $\bar{I}_{S,min}$). Color code of atomic models: Ni = gray, Fe = orange, O = red, Alkali = blue, H = white.

somewhat dependent on the experimental conditions, and decrease to ca. −40 mV pH$^{-1}$ at slower scan-rates or at higher concentrations (Supplementary Fig. 18a–c). This is typical when mass transfer limitations are involved, as also inferred by the response of the peak separation to these parameters (Supplementary Fig. 18d, g). Since the uncompensated solution resistance ($R_u$, iR-drop) increases at low concentration/pH, this might indeed affect the kinetics (see Supplementary Table 8). More unexpected, the cathodic redox peak ($E_{p,c}$) is less affected by the electrolyte pH than the anodic peak ($E_{p,a}$), and actually exhibits a close to Nernstian pH-slope (ca. 0 mV pH$^{-1}$ in the RHE scale) (see Discussion, Supplementary Fig. 18b, c). The absolute level of redox charge ($Q_{peak}$) in both the oxidation/reduction waves is on the other hand not that greatly affected by the cation or pH as the peak potential in accord with the XAS data above (Supplementary Fig. 18e–i), yet, both higher pH and slower scan-rates help approach the unity value of ~1.8 e$^-$ per Ni$^{-1}$.

**Activity descriptors for alkali metal cations investigated using DFT.** DFT calculations were performed of bulk NiOOH and of a (100) surface slab model of both NiOOH and Fe-doped Ni(Fe)OOH in the presence of hydrated alkali metal cations (Fig. 5 and Supplementary Fig. 19). The full details of the DFT method are provided in the Methods section and Supplementary Note 5.

Our results are consistent with an increase in the cation-oxygen (X-O) coordination distances from Li$^+$ to Cs$^+$, both to nearby water-associated oxygens and to lattice oxygens belonging to Ni(Fe)OOH (Supplementary Fig. 19 and Supplementary Table 9). The increase in the X-O distance amounts ca. 4.5% from Li$^+$ to Cs$^+$, in agreement with Zaffran et al.[7] This also causes an increase in the layer separation from Li$^+$ to Cs$^+$. Further insight regarding the catalytic activity of the alkali cation-Ni/FeOOH system was obtained from electronic structure calculations. Bader charge

analysis is a standard method for assessing charge distribution within a compound[61]. The charges of both the Ni- and Fe-sites remain essentially unaffected regardless of the alkali cation (Supplementary Fig. 20). Based on our previous knowledge, the cation should not be modifying the catalytic activity of Ni(Fe) OOH significantly, however, below we will more carefully evaluate the catalytic activity using a set of local reactivity properties that correlate with the Lewis basicity/acidity of a metal site. The OER reaction path has recently been shown to pass through a bifunctional mechanism relying on the cooperation between a neighboring Ni–O Lewis acid-base pair[30]. This justifies that improved information of the reactivity of the Ni(Fe)OOH surface can be obtained by estimations of the local Lewis acidity and basicity of Ni, Fe and O lattice sites as a function of the intercalating cation. For this purpose, we have employed three local reactivity properties (estimated by DFT calculations) that have previously been shown to correlate with the acidity/basicity of surface sites[44–46]. We model undercoordinated "edge sites" since these have been established as the more reactive sites in the Ni–Fe catalyst (details on the model is provided in Supplementary Fig. 21)[31,33–35,60]. Figure 5 shows the variation in these properties for the Fe-doped version of a γ-Ni(Fe)OOH(100) surface, often employed as a model for the catalytic behavior of Ni–Fe catalysts[28,30]. The maximum value in the surface electrostatic potential, $V_{S,max}$, at the Ni site gives a measure of the electrostatic contribution to the local Lewis acidity of the metal site, whereas the minimum of the surface local electron attachment energy, $E_{S,min}$, reflects the charge-transfer capacity of this site. Similarly, minima in the surface electrostatic potential and the surface local average ionisation energy, $V_{S,min}$ and $\bar{I}_{S,min}$, respectively, indicate the electrostatic and charge-transfer contribution to the Lewis basicity of the oxygen site. We find, in agreement with the lack of a significant variation in the Ni and Fe Bader charge, that these properties show no significant variation

with the intercalating electrolyte cation (Supplementary Table 10). The exception is the $E_{S,min}$ value, which indicates a slight increase in the Lewis acidity of the Ni site from small to large cation ($Li^+$ to $Cs^+$). As will be discussed in the next paragraph, this "activation" by the alkali metal cations is relatively small in relation to the effect of, e.g., substituting Ni for Fe in the NiOOH lattice, so the magnitude of change from $Li^+$ to $Cs^+$ supports only a minor change in the reactivity descriptors. The Lewis basicity of the oxygen sites also remains largely unaffected by the cations (estimated from the $V_{S,min}$ and $\bar{I}_{S,min}$ values in Supplementary Table 10). Thus, the reactivity indicators suggest that the alkali cations have only a minor direct promoting effect on the catalytic activity of NiOOH.

We further model the Fe-doped Ni(Fe)(OOH) catalyst. When a Ni atom is exchanged for an Fe atom, the Lewis acidity of the nearby Ni-site increases (Fig. 5d–h and Supplementary Table 11). This can be exemplified using $K^+$ as an intercalating cation. The Ni atom next to an Fe-site becomes "activated" (i.e. stronger Lewis acid) whereas the neighboring O sites become slightly "deactivated" (i.e. less Lewis basic = more positive $V_{S,min}$). This is indicated by the change in the local reactivity properties, both in the charge-transfer and electrostatic components of the Ni-site, and the electrostatic component of nearby O-sites ($V_{s,min}$). The change in Lewis acidity (cf. $E_{S,min}$) of the Ni-site is of the same magnitude or only slightly larger than the promoting effect from $Li^+$ to $Cs^+$. On the contrary, the Fe-site is significantly more Lewis acidic compared to the Ni-site, despite the activation of the Ni-site. This is in line with recent work by Nocera and coworkers where $Fe^{3+}$ was argued to be the stronger Lewis acid and predicted to increase the acidity of the aqua/hydroxo group connected to the Ni-site[24], which was also confirmed with situ Raman spectroscopy by Edvinsson and co-workers[62]. We also find that the oxygen sites connected to an Fe-site are significantly less Lewis basic than the oxygens connected to a Ni-site. The most negative charge-transfer capacity ($E_{S,min}$ value) is, not surprisingly, found for the sites occupied by Fe atoms, suggesting that these sites are most reactive of the two. The $V_{S,max}$ value (i.e., electrostatic component) also changes in line with an increased Lewis acidity. Our results thus point toward Fe being the most Lewis acidic (and thus electrophilic) site along with less basic oxygen ligands, which could facilitate a prospect nucleophilic attack on this site. We do not find any differences in how the cations influence either Ni or Fe-sites, so we conclude that both sites are equally affected by alkali metal cations, which is significantly less compared to the effect of lattice Fe substitution. Altogether, our analysis indicates that Fe-dopants have both a direct and indirect effect on the local reactivity properties, and thus of the general reactivity of the Ni(Fe)OOH catalyst, whereas the alkali metal cations have only a comparative minor promoting effect on the OER activity.

## Discussion

Several hypotheses behind the impact of electrolyte cations on the OER activity have been put forward; (I) strong non-covalent interactions between small alkali metal cations and chemisorbed species forming cation-$OH_{ads}$ clusters that block the active sites[10]. (II) perturbation of adsorption energies of the OER intermediates and the kinetics[7,50]. (III) promotion of peroxo-like "active oxygen" species by larger cations leading to a higher activity[8,40]. Here we put forward a new hypothesis: (IV) a change in the electrolyte pH caused by the differences in the Lewis acidity of the alkali cations (and thus the basicity of the alkali hydroxides) modifies the OER activity.

Our XAS and electrochemical data support that electrolyte cations affect both the $Ni^{2+/(3+\delta)+}$ redox process and the OER activity, where large cations shift the oxidation and the OER process to lower potentials. This is explained by an increase in the electrolyte pH in the presence of larger cations due to the higher

basicity of the corresponding alkali hydroxide. In recent findings by Pasquini et al.[63] the catalytic rate of a $CoO_x$ catalyst was found to be controlled by the levels of oxidized $Co^{4+}$ at any defined OER overpotential. A similar scenario could prevail in our Ni–Fe catalyst, although, we do not find as strong correlation with the population of oxidized $Ni^{(3+\delta)+}$ during steady-state conditions and the OER activity. We also cannot exclude a correlation between the OER activity and the redox-process due to an increase in the through-film conductivity upon formation of $Ni^{3+/4+}$ as shown by Boettcher and co-workers[47].

The metal oxidation step in the Ni–Fe catalyst occurs via a deprotonation step according to Eq. (1).

$$Ni(OH)_2 + OH^- \leftrightarrow NiOOH + e^- + H_2O \qquad (1)$$

It has been demonstrated that the activity and redox-process of the Ni–Fe catalysts are sensitive to the electrolyte pH[48]. Both the OER activity and anodic redox-peak exhibit pH slopes that deviate from Nernstian[64] (i.e., −40 to −90 mV pH$^{-1}$ on the RHE scale). Earlier studies have attributed this to non-concerted proton-electron-transfer pathways, reported as ~2 protons transferred per 1 electron (H$^+$/e$^-$ ratio of ~2)[40,48]. This might be explained in accordance with anionic redox-activity on the oxygen ligands as seen in soft XAS[36,37], and/or by the formation of peroxo-like (*O$^-$ or *O$_2^-$) intermediates denoted as "active oxygen", seen in Raman spectroscopy as a band between ~800–1150 cm$^{-1}$[38,39]. Concerning electrolyte cations, Garcia et al.[8] found a higher intensity of this Raman band in CsOH compared to LiOH, thus large cations were proposed to promote "active oxygen". We note that in earlier studies from their group this band was shown to intensify with increasing electrolyte pH[38,39]. Since we now know that larger electrolyte cations (i.e. $Cs^+$) increases the electrolyte pH, we argue that it could explain why large alkali metal cations promote the "active oxygen" band. The nature of these reactive species could in turn explain why a higher pH (induced by the presence of large cations) promote a higher OER activity. The non-Nernstian pH-dependence of the cathodic redox-peak (0 mV pH$^{-1}$ on the RHE-scale) has already been explained by Merrill et al.[41], where NiOOH was found to follow a different charge and discharge mechanism. The "active oxygen" species formed during charge, vanish before reaching the onset of the reduction wave, whereby the discharge occurrs via a $NiO(H_2O)$ intermediate before reduced back to $Ni(OH)_2$, hence explaining the difference in pH-slopes between the oxidation/reduction waves.

Other considerations regarding electrolyte cations concern the water solvation shell, which is more strongly held around smaller cations ($Li^+$) compared to larger cations ($Cs^+$)[17], whereby the native H-bonded network of water is disrupted to a higher extent in the presence of smaller cations. This has implications on the diffusion rate and water reorientation times, which is usually slower in the presence of small cations[14,65], which also affects the proton-transfer kinetics[66]. In a study by Ledezma-Yanez et al.[67] a correlation between the interfacial water reorganization and the activity for the hydrogen evolution reaction (HER) on metallic Pt was established, where a faster water reorganization was found to benefit a higher HER activity via more efficeint proton/hydroxide transfer through the electric double layer. Huang et al.[18] showed that easier water reorganization around a redox-active metal center such as ferri/ferrocyanide in the presence of large cations, is related to a higher exchange current density and a lower reaction entropy and to faster electron transfer kinetics. We therefore cannot exclude that there are some specific cation effects in addition to the pH effects due to non-covalent interactions between cations and water.

Our DFT-derived reactivity descriptors show that alkali metal cations have a rather small effect on the local reactivity properties

of the Ni/FeOOH surface compared to e.g. Fe-substitution, which activates Ni-sites by increasing the Lewis acidity. Despite this, the Fe-sites remain the most Lewis acidic site (and thus the most "reactive" site). In contrast to the rather large effect of Fe-substitution on the reactivity descriptors in comparison to the alkali cations, there is still a small increase in the acidity for both Ni and Fe sites between the two extremes (Li+ to Cs+). Since this effect is relatively small, it is likely to go unnoticed unless it exceeds the effect of the electrolyte pH. In catalysis, it should also be kept in mind that "optimal" activity is often associated with a moderate reactivity (cf. sweet spot in adsorption energy in a Volcano plot).

In conclusion, we demonstrate that the effect of alkali metal cations on the OER activity of the $Ni_{65}Fe_{35}(OOH)$ catalyst can be explained by differenes in the electrolyte pH, supported by the directional shift of the $Ni^{2+} \rightarrow Ni^{3+/4+}$ redox-peak and the OER activity. Since the basicity increases in the order LiOH < NaOH < KOH < RbOH < CsOH following the cation size (explained by their $pK_b$ values), the electrolyte pH increases accordingly from small to large alkali metal cations due to an increase in the amount of dissociated $OH^-$. Since the OER activity is sensitive to pH, this is able to explain previously puzzling discrepancies between the trend in cation size and the OER activity, since any factor introducing uncertainties in the $OH^-$ concentration (or electrolyte concentration) can affect the OER activity. Our DFT-derived reactivity descriptors further support that alkali metal cations do not significantly alter the intrinsic reactivity properties of either Ni, Fe, or O lattice in contrast to Fe substitution, which instead increases the Lewis acidity of neighboring Ni-sites (although Fe remains the most Lewis acidic site). The alkali cations are therefore unlikely to account for the differences in the OER activity. Thus, our DFT results are in line with the experimental findings that the electrolyte pH constitutes the strongest activity descriptor. Future studies therefore need to explore possible pH-effects in oxide-derived catalysts in striving for understanding the role of alkali metal cations in OER electrocatalysis.

## Methods

**Ni–Fe catalyst depostition and electrolyte preparation.** The Ni–Fe(OOH) catalyst was prepared by electrodeposition on ~25 mm thick conductive graphene sheets (Graphene Supermarket) roughened with sand paper; without this step, the adhesion and stability of the films were poor on the smooth surface. The deposition solution constituted 9 mL of 50 mM $Ni(NO_3)_2 \cdot 6H_2O$ (99.999% trace metals basis, Sigma-Aldrich) and 1 mL of 50 mM $Fe(NO_3)_3 \cdot 9H_2O$ (≥99.999% trace metals basis, Sigma-Aldrich); Fe tends to deposit easier than Ni. The catalyst films were deposited galvanostatically on a geometric area of ca. 0.4 $cm^2$ using a 2-electrode setup where a fine Pt-coil served as counter electrode. A current density of $-2.5$ mA $cm^{-2}$ was applied for 113 s, using a Bio-Logic SP-200 potentiostat. This resulted in a film with a Ni:Fe stoichiometry of 65:35 at. %, and a geometric metal loading (with respect to both Ni and Fe content) of ~25 ± 2 μg $cm^{-2}$. Another $Ni_{75}Fe_{25}$ catalyst with a metal loading of 3 ± 1 μg $cm^{-2}$ (~50 nm thick) was instead prepared from 10 mM deposition solutions including 100 mM $KNO_3$ supporting electrolyte. The alkali metal hydroxides used for activity investigations were the following; LiOH (monohydrate, 99.995% trace metals basis, Sigma-Aldrich), NaOH (monohydrate ≥ 99.996% metals basis, Alfa-Aesar), KOH (semiconductor grade, pellets, 99.99%, Sigma-Aldrich), RbOH (hydrate, Sigma-Aldrich), CsOH (monohydrate, 99.95% trace metals basis, Sigma-Aldrich). Unless otherwise stated, the alkali hydroxides were stripped of Fe-impurities according to a modified purification method reported by Boettcher and co-workers (Supplementary Note 1)[47].

**In situ XAS at the influence of alkali metal cations.** XAS was acquired at the Ni and Fe K-edges at the P64 beamline at Petra III (DESY, Hamburg, Germany) during in situ conditions. The $Ni_{65}Fe_{35}(OOH)$ catalyst was investigated in purified 0.1 M alkali hydroxides; LiOH, NaOH, KOH, RbOH, and CsOH. The electrochemical cell constituted a single compartment 3-electrode cell, with an outer diameter of 50 mm, an inner diameter of 25 mm, and an X-ray window of 6 × 6 mm mounted ~20 mm from the bottom. The working electrode was an Ni–Fe (OOH)/graphene sheet connected via a Cu-clip from the upper backside, a Pt-mesh served as counter electrode, and a leak-free Ag/AgCl as reference electrode. All measurements were controlled using a Bio-Logic SP-200 potentiostat. Spectra (XANES/EXAFS) were collected at different potentials; 1.1, 1.34, 1.41, 1.48, 1.56, 1.66 V and return to 1.1 V vs. RHE. The uncompensated solution resistance ($R_u$, iR-drop) was determined individually using electrochemical impedance

spectroscopy (EIS) at 10 kHz, applying 85% automatic iR-compensation (ZIR method) in the EC-Lab software. Additional 5% was compensated afterwards to total level of 90%. More than 90% was not possible due to overcompensation effects. A typical value of the iR-drop in the in situ XAS setup was 25 Ohm in 0.1 M electrolytes. The spectra at the Ni and Fe K-edges were collected in pairs before stepping to the next potential, where typically 2–5 consecutive spectra were collected for each edge. The incoming X-ray beam was set to an incident angle of ~10°, and the fluorescence collected from the back-side using a passivated implanted planar silicon (PIPS) detector. A second data set was collected at the Ni K-edges at the SuperXAS-X10DA beamline at Swiss Light Source (SLS) (PSI, Villigen, Switzerland), and is shown in Supplementary Fig. 11. For these measurements, a Si-drift detector with five crystals was used instead. More details on the XAS data processing and EXAFS simulations are provided in Supplementary Note 2. All electrode potentials were converted to the RHE scale afterwards, where the offsets were obtained by calibration against an RHE electrode.

**Correlations between OER activity and electrolyte pH measured "in house".** All "in house" electrochemical characterization was carried out in 3-electrode configuration using the same electrochemical cell as used for in situ XAS. The cell setup differed mainly in the choice of the reference electrode, which was a RHE electrode in order to avoid unknown shifts in the potential-scale due to small differences in pH between the alkali hydroxides, and we applied constant purging with $N_2$. The employed RHE electrode was manufactured from a hollow glass rod (∅ = 8 mm) with a coiled Pt-wire inside. The rod was sealed at the top around the Pt-wire using a hot flame. The cavity of the RHE was filled with the respective alkali hydroxides (LiOH, NaOH, KOH, RbOH, CsOH) that had been degassed with $N_2$ for ~10 min to remove $O_2$. Hydrogen ($H_2$) was evolved on the inner Pt-coil by applying $-5$ V for about ~30–60 s. The OER activity was investigated in several electrolyte concentrations; 0.05 M, 0.1 M, 0.25 M, 0.5 M, and 1 M of LiOH, NaOH, KOH, RbOH, and CsOH. The alkali hydroxides had been purified unless otherwise stated (see Supplementary Note 1). Fresh catalyst films were always used for different cations, as well as for repetitions of the same cation. The electrolyte pH was determined using a standard pH-meter pre-calibrated between pH 9–13 (pH-electrode LE438, Mettler Toledo). For some measurements, a specialized alkali pH-electrode with a ceramic membrane was employed (InLab Routine® Pro-ISM, Mettler Toledo). The pH of LiOH could not be determined with any of these pH-electrodes since the small $Li^+$ ion seems to have too similar transport properties to the proton. The experimental pH of LiOH was instead estimated using the potential difference between the RHE electrode and the Ag/AgCl reference electrodes, by constructing a calibration curve using commercial solutions with known pH (Supplementary Note 3). Tafel slopes were collected during steady-state conditions by applying chronoamperometric steps for 250–300 s, increasing by 5–50 mV per step. The first 120 s were discarded from the average to assure steady-state conditions. Automatic iR-compensation was applied to all potentiodynamic and potentiostatic techniques to a total level of 90%; 85% automatic, and 5% manual compensation.

**Elemental analysis.** Inductively coupled plasma optical emission spectroscopy (ICP-OES) was used to determine the metal loading and compositions in the Ni–Fe catalyst and the concentration of alkali cations in the electrolytes, using a Thermo Fisher Scientific iCAP 6000-series spectrometer. The Ni–Fe catalyst films were dissolved in 200 μl $HNO_3$ (70%), 200 μl $H_2SO_4$ (>97.5%), and 500 μl HCl (37%) (all ACS reagent grade, Sigma-Aldrich). The samples were left for 1–2 h to digest into its elements, and then diluted to a total volume of 5 mL using ultrapure MilliQ-water (18 MΩ cm). The alkali hydroxides were instead diluted with MilliQ-water to a final concentration of 12.5 mg/L. The dilution factors for the 1 M hydroxides were following: LiOH (3356x), NaOH (4641x), KOH (4489x), and RbOH (8198x), and CsOH (13 434x) and the 0.1 M solutions were diluted 10 times less. Low concentrations minimize the easy ionizable effect (EIE). ICP standards were prepared from a multielement standard solution (100 mg/L, TraceCERT, Sigma-Aldrich), and diluted in a series of 0.05, 0.1, 0.5, 1, 10, and 20 mg/L. Note that the concentration of Cs in CsOH was not successfully determined because of near zero-emission intensity. To scan for trace impurities, the alkali hydroxides were kept more concentrated (~1000 mg/L). We assume that only ionic impurities in the solution is of importance. The analysis was carried out using an Avio 200 spectrometer (PerkinElmer), and a multielement standard solution with a concentration of 10 mg/L (PerkinElmer) used without further dilution. The following emission wavelengths were analyzed; Li (460.2 nm, 670.8 nm), Na (589.6 nm), K (766.5 nm, 769.8 nm), Rb (780.0 nm), Cs (672.3 nm, 455.5 nm), Ni (231.6 nm), and Fe (238.2 nm). The hydroxides were also scanned for additional impurities; Ca (393.3 nm), Mn (257.6 nm), Co (228.6 nm), Cu (327.4 nm), and Zn (206.2 nm).

**Scanning electron microscopy (SEM) and energy-dispersive X-ray spectroscopy (EDS).** SEM was carried out using a JSM-7000F microscope (JEOL). Additional SEM and elemental mappings were carried out using a LEO1550 microscopy (Zeiss). For images, the working distance was set between 5.5–10 mm and the accelerating voltage to 3–10 kV depending on the information depth and the microscope. To determine the elemental compositions, EDS was carried out at an accelerating voltage of 15–20 kV using the built-in EDS detector in the

respective microscopes (both from Oxford Instruments). The analysis was carried out using either the INCA or AZtec software packages (Oxford Instruments).

**DFT computational details.** Atomic models of the alkali cation impregnated NiOOH bulk material were generated from the $Na^+$-containing structures optimized by Zaffran et al.[7]. The NiOOH and Fe-substituted Ni(Fe)OOH has a layered structure, with $H_2O$ and alkali cations residing in-between the crystal Ni/FeOOH sheets (Supplementary Figs. 19 and 21). The bulk parameters were reoptimized for the full series of alkali metal cations from $Li^+$ to $Cs^+$ (and $Fr^+$). Surface slab models of the (100) facet, often used as a representative surface for the Ni/FeOOH material, were constructed from the optimized bulk material. Four layers of Ni/FeOOH were used in the slab models employing a $(1 \times 1)$ surface supercell, and a vacuum distance of 17 Å. Only the top layer was allowed to relax in the calculations. The surface exposes three $Ni^{3+/4+}$ ions. For comparison, we exchanged one third of these with $Fe^{3+/4+}$ at different positions in the analysis of surface properties (vide infra). Periodic and spin-polarized DFT calculations were performed with the Vienna Ab-initio Simulation Package (VASP)[68]. The PBE exchange-correlation functional was used throughout, employing the Hubbard+U corrections[69] for the Ni and Fe 3d states with the U−j values 6.0 and 3.5 eV, respectively, as suggested by Zaffran et al.[7]. The core states were represented by standard PAW potentials[70,71], whereas the valence electrons (Li: 1 s 2 s; alkali[p=period]: (p-1)s (p-1)p ps; Ni/Fe: 3p 3d 4 s; O: 2 s 2p; H: 1 s) where expanded on a plane-wave basis with an energy cut-off of 800 eV for the bulk calculations and 600 eV for the surface slabs. The **k**-space was sampled using the tetrahedron method with Blöchl corrections[72] on $8 \times 6 \times 4$ and $4 \times 6 \times 1$ **k**-meshes for the bulk and surface calculations, respectively. In the geometrical optimizations, the forces were relaxed to <0.01 eV/Å, while the electronic convergence criterion was set to $1e^{-6}$ eV. More details on the specific DFT-derived reactivity descriptors are found in Supplementary Note 5. The robustness of the computed results was tested by evaluating the effect of including empirical dispersion corrections via the D3(BJ) method[73,74], and implicit solvation following the VASPSOL protocol[75]. The effect of dispersion corrections on the predicted reactivates is found to be negligible (Supplementary Tables 12 and 13). Whereas inclusion of solvation is found to alter the absolute values of the local reactivity properties, the relative trend over the series of cations remains the same with or without solvation (Supplementary Table 14).

## Data availability

All data in this manuscript can be made available upon request to the corresponding authors.

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

## Acknowledgements

We thank the Knut and Alice Wallenberg Foundation, the Swedish Research Council, and the Åforsk Foundation, and StandUp for Energy for financial support. We also thank Kjell Jansson at Stockholm University for assistance with SEM-EDS measurements. We also greatly thank Wolfgang Caliebe and the staff at the P-64 beamline at Petra III (DESY, Hamburg), with financial support from the Bundesministerium für Bildung und Forschung (BMBF) project no. 05K19PXA. We thank Urs Vogelsang of the SuperXAS beamline at Swiss Light Source (PSI), and Juan Herranz Salaner at PSI for support. Computational resources were provided by the Swedish National Infrastructure for Computing (SNIC) at the National Supercomputer Center (NSC).

## Author contributions

M.G. carried out the data analysis, wrote the manuscript and prepared the figures for the experimental data results. J.H.S. carried out the density functional theory calculations and prepared the figures for the DFT data results, and wrote the text for the DFT section. O.D.M. contributed specifically to data interpretation, overall experimental design, and electrochemical characterization. M.G., O.D.M., S.K., H.Y.W., M. Börner, and M.S. all contributed equally to XAS data collection, and S.K. and H.Y.W. contributed especially to XAS data interpretation. A.K., V.M., O.V.S., M.N., and M. Bauer contributed to XAS data collection and overall experimental design/setup, J.D. and A.U. assisted with elemental analysis, and A.N. provided hands-on supervision and support throughout the project. The manuscript content has been commented and discussed by all co-authors.

## Funding

## Competing interests

The authors declare no competing interests.
