## [Peer Review File · Nature Communications]

REVIEWER COMMENTS

Reviewer #1 (Remarks to the Author):

The manuscript by Gorlin et al. discovered that the alkaline metal cation effect on the OER activity of NiFe oxyhydroxide catalyst was misinterpreted in previous studies. The observed OER catalytic activity trend in different alkali hydroxides is actually due to the inaccurate determination of the pH values of the electrolytes, instead of a specific alkaline metal cation effect. It is not surprising that the pH values have the most significant influence on the reported OER activity: not only the OH⁻ concentrations are different but also the reported potential values, which are commonly converted using the Nernst equation, are incorrect. This finding is important as it may clarify some of the misunderstandings in the field and in theory suited to Nature Communications. However, substantial revisions are still needed to address the following points before publication:

1) A more straightforward way to demonstrate that the alkaline metal cations have no effect on the OER activity is to measure the OER activity of the catalyst in different alkali hydroxides at the same experimental pH. The reproducibility of the catalysts used for the measurements should be first confirmed: either samples with identical OER activity in the same electrolyte can be made or the same sample can be used repetitively after extensive rinsing without degradation. The LiOH may be spared from the experiment due to the difficulty in determining the pH value experimentally. Preferably more than one pH value should be tested.

2) The latter part of the manuscript seems to be more concerned about whether Ni or Fe is the active center in the NiFe oxyhydroxides than the influence of alkaline metal cations. The authors' argument follows the hypothesis that Ni is the active center while Fe is a Lewis acid that enhances Ni activity. Some researchers have challenged this hypothesis by arguing that the Fe is more than a Lewis acid and is actually the active center of the NiFe oxyhydroxides (<https://doi.org/10.3390/molecules23040903>; <https://doi.org/10.1016/j.joule.2018.01.008>). The XAS data and DFT calculation presented in this manuscript do not show further evidence (over previous studies) on Ni being the active center. For example, the XAS data does not prove Ni is oxidized to Ni(IV). Furthermore, despite DFT calculation showed that Fe "edge sites" were the most reactive sites, the authors claimed that the reactivity might too high that they are not catalytically active. But this is purely speculation. I suggest that the authors do not focus on the identification of the active site if they don't have solid evidences. Instead, they should focus on demonstrating that alkaline metal cations have no influence on the NiFe oxyhydroxides, whether Ni or Fe is the active center.

3) The data in the manuscript may need to be analyzed with more care. For example, it is claimed that there is a strong correlation between the Ni²⁺/Ni³⁺ redox peak potential and the OER potential because there is a constant separation between the two processes (Figure 4) and both exhibit near-identical slopes vs pH (Figure 5c). However, when looking carefully into the data in Figure 4, the potential separations are not as "constant" as the authors claimed. The slope fitting for OER in

Figure 5c also seems not to include all the data points. Since direct conclusions are drawn from these analyses, it should be taken with great caution. There is also a previous study by Boettcher's group which demonstrated that the Ni²⁺/Ni³⁺ redox peak potential did not correlate with water oxidation activity (<https://doi.org/10.1021/acscatal.5b02924>).

4) The "exceptions" observed in the data should also be considered more carefully. In analyzing the Ni K-edge shifts in different electrolytes, the shift increases in the order similar to the activity trend but with an "exception". However, one might also argue with this "exception" that Ni K-edge shift is not a real indicator of OER activity.

5) Regarding the iR correction, was the resistance (R) of every sample measured independently? If so, the experimental details for determine the resistances should be provided. If not, it is highly recommended to do so since they may be different in different electrolytes.

6) Regarding the unexpected drop in activity in RbOH, have the authors tried a different batch of RbOH since they suspected it might have larger carbonates or water contaminations?

7) Some other minor issues:

Page 1: metal-air batteries are not relevant to the discussion of H₂ production from water.

Page 3: the TOF value in LiOH should be 0.04.

Page 4: the compositional inhomogeneity shown in Figure S3 and Table S2 is not "small".

Page 5: Figure S5b should be Figure 5b.

Figures S6, S7: labeling errors.

Reviewer #2 (Remarks to the Author):

Comments and Suggestions for Authors of: NCOMMS-20-10153

General comments:

The authors present a study of key activity descriptors for the OER reaction on Ni-Fe oxyhydroxide catalysts and conclude that pH is the dominating one, rather than a specific cation effect. The manuscript is largely well written, and I agree with the authors conclusions. For the field of electrolysis, it is important to clarify the role of different descriptors. However, there are a number

of points, especially in the practical work where the authors need to specify, explain, and motivate better. On a few points it might also require some additional analysis or measurements. For many of the parameters presented, the difference between the different cations are within or close to the error margin, thus it becomes very important to show that the authors really have done the practical work and the analysis correctly.

Major Issues

1. There must be a typo in the numbers regarding results of TOF where it says: "The intrinsic activity was evaluated as TOF_{mass} (turnover frequency per total metal ion loading), which justified the geometric activity trends (Figure 2c), with variations between 0.4 s⁻¹ (LiOH) to 0.1 s⁻¹ (CsOH) (see Table S1)". From fig 2c and table S1 it should be 0.04 s⁻¹?
2. I am missing information and discussion on the purity of the different electrolytes. The authors present a rigid method employed for electrolyte purification in the SI, but I don't find information on the purity of the starting chemicals. It would also be good to measure or estimate the types and concentrations of impurities in the electrolytes used. As the differences between the different solutions are so small, and this type electrochemical reactions are highly sensitive to contaminations, this information is of high importance.
3. The iR correction needs to be explained better, and possibly redone. The authors state that measurements were done with "with 85 % iR-correction" and that "A typical value of the iR-drop in the in situ XAS setup was 25 Ohm". I have two questions here: 1) What happened with the remaining 15% iR-loss? Was that left uncompensated? If so, that should have a marked effect on the results of both pH and cation effect? 2) it appears strange to give one typical value for the uncompensated resistance when the alkali concentration changed almost two orders of magnitude, this ought to affect the uncompensated resistance a lot?

It would be nice to see a table in SI with the values of the uncompensated resistance in all measurements, especially the different alkali concentrations.

4. The authors should motivate and explain more on the pH measurements and how valid these are. From the text and Table S4 it is clear that no results were obtained for LiOH due to "Erroneous readings". Why was this happening? Could the authors not use another instrument or another technique? Additionally, in my experience, a "standard pH-meter" as the authors seem to use are often decently accurate around neutral pH, but for highly acidic or highly alkaline solutions the accuracy is often quite bad. Since pH is a crucial parameter in this work, I think more care and more precision should go into determining it.
5. In the Results section it is stated that "A home manufactured reversible hydrogen electrode (RHE) was employed as reference electrode to avoid unknown potential-shifts due to the pH". But in the Methods section it says that "a leak-free Ag/AgCl employed as reference electrode" and that the "offsets were determined by calibration against an RHE electrode". This needs some clarification as it is quite a different thing to use an RHE electrode as reference electrode and to calibrate a Ag/AgCl electrode.

Minor Issues

1. First sentence in the introduction states that it is imperative to “slow-down the steadily growing carbon footprint”, it should be the growth that should be slowed down, the footprint does not really have a motion.

Reviewer #4 (Remarks to the Author):

Currently, serious energy and environmental crises are the grand challenges facing society worldwide. The correlative investigations of electrocatalysts for water splitting process (e.g. OER and HER) have become one of the hottest areas. In this work, the authors investigate the OER activity of the Ni-Fe oxyhydroxide, where the activity increases in the order of $\text{LiOH} < \text{RbOH} < \text{KOH} \approx \text{NaOH} < \text{CsOH}$. They claim that the electrolyte pH (rather than the cation size) can play an important role in affecting the OER activity, and therefore the electrolyte pH can serve as an activity descriptor to determine the OER activity. This work is interesting, however, the following issues should be addressed:

- 1) From figs 7C and S16, it can be found that many hydrogen bonds exist in these studied structures. Therefore, a semi-empirical van der Waals (vdW) correction must be employed to account for the correlative dispersion interactions for this kind of systems. However, it cannot be judged whether the vdW correction has been added during the computational process. Thus, authors must clarify it in view of the reliability of computational results.
- 2) The authors should consider the effect of solvation on the correlative computed results, in view of the fact that the OER reaction occurs in water as a solvent
- 3) For the convenient understanding, the authors should clarify the definition of “Fe edge site” in detail.

Author's Response to Reviewer #1:

The manuscript by Gorlin et al. discovered that the alkaline metal cation effect on the OER activity of NiFe oxyhydroxide catalyst was misinterpreted in previous studies. The observed OER catalytic activity trend in different alkali hydroxides is actually due to the inaccurate determination of the pH values of the electrolytes, instead of a specific alkaline metal cation effect. It is not surprising that the pH values have the most significant influence on the reported OER activity: not only the OH⁻ concentrations are different but also the reported potential values, which are commonly converted using the Nernst equation, are incorrect. This finding is important as it may clarify some of the misunderstandings in the field and in theory suited to Nature Communications. However, substantial revisions are still needed to address the following points before publication:

First, we thank the reviewer for the suggestions. We have made major revisions to address the reviewer's comments, and new data has been added to the manuscript to further support our conclusions.

1) A more straightforward way to demonstrate that the alkaline metal cations have no effect on the OER activity is to measure the OER activity of the catalyst in different alkali hydroxides at the same experimental pH. The reproducibility of the catalysts used for the measurements should be first confirmed: either samples with identical OER activity in the same electrolyte can be made or the same sample can be used repetitively after extensive rinsing without degradation. The LiOH may be spared from the experiment due to the difficulty in determining the pH value experimentally. Preferably more than one pH value should be tested.

We agree with the reviewer's point that the reproducibility is important, and we like the idea of cross-checking the activity in electrolytes where the pH has been adjusted to similar values. Regarding the reproducibility of the electrochemical OER activity measurements, each data point reported in our manuscript is the average of 2-5 repetitions and the error bars correspond to the standard error. Figure R1 shows the reproducibility of our measurements, by comparing compares the cyclic voltammograms (CVs) from several different catalyst films in 0.1 M alkali hydroxides. We did not stir the electrolyte during these measurements; however, we did have a constant flow of N₂ gas which introduce some convection. Therefore, we did not focus on stability in measurements, yet, the short-term stability appears to be good within the time frame of the measurements (see Figure R1b). The differences observable in Figure R1a are due to small variations in the metal loading of the different films, as well as partial electrode blockage by O₂ bubbles (especially after the film being exposed to high potentials where high current densities are evolved), as demonstrated in Figure R1c.

Figure R1. (a) Reproducibility of the Ni₆₅Fe₃₅ catalyst CVs in purified 0.1 M alkali hydroxide electrolytes; CVs at a scan-rate of 10 mV/s. (b) Short-term stability of the Ni-Fe film in KOH before and after Tafel slope collection.

In attention to the reviewer's suggestion, we are now providing data where the electrolyte's pH has been adjusted to nearly identical values (or at least similar values) while changing the cation (see Figure R2). The experiments were carried out in three different adjusted pH-values; ~13.00, ~13.45, and pH ~13.79, corresponding to ~0.1 M, ~0.25 M, and ~0.5 M concentrations of (non-purified) alkali hydroxides. We did not leave out LiOH since it is good as a reference point and to guide the eye, however, since the pH cannot be determined very easily and with high accuracy in LiOH, the values were omitted from the pH-trend. We first cross-checked the activities in the non-adjusted electrolytes (so with the same concentration) to make sure we could recover the initial characteristic "cation trends". In the electrolytes where the pH-values have been adjusted close to identical values, the new OER activity values indeed align more closely on top of each other, or more importantly, the old characteristic cation trend is displaced. The results are presented in Figure R2c.

Please note that there are significant differences in the measurement protocol; we used freshly prepared electrolytes (and non-purified) which came from new batches from the supplier (we ran out of the old ones). We also used a "cation switching" protocol to avoid discrepancies from differences in the metal loading since this would otherwise require a substantial number of repetitions (i.e we used the same catalyst films for pH 13.00/0.1 M, pH 13.45/0.25 M, and pH 13.79/0.5 M). We thoroughly rinsed the films with MilliQ-water in between the cations to avoid cross contamination. To demonstrate the effective cleaning protocol, we show the OER activity can be recovered upon switching in the order of CsOH-LiOH-CsOH (see Figure R2g). We did not measure the cations in a specific order, but they were scrambled to make sure there was no ageing or "memory" by exposing the films to e.g. small or large cations. We also used a new special type of pH-electrode, which has a ceramic membrane and should minimize the alkali error (InLab[®] RoutinePro-ISM, Mettler-Toledo). Despite this, we encountered the same problem in LiOH as with the old pH-electrode, and we could not determine the pH in LiOH (the pH decreases with increasing concentration). This means that the alkali electrode does not eliminate the transport problems over the membrane due to the small size of the Li⁺ cation. We also stress that the adjusted pH-values are not perfectly identical, we found this to be challenging with the pH-meter. Also, the difference in the concentration required to reach a similar pH-value, we believe might introduce other non-linearities between the cations since the iR-drops instead are different.

The measurements in Figure R2 are now presented in the Supporting Information of the manuscript as Figure S17, and the following paragraph was added to the main text on p. 8 of main manuscript;

The pH effect was further verified by experiments where the pH was adjusted to near identical values in all the alkali hydroxides. This shows that the “characteristic” cation trend is easily displaced by changes in the pH, and the OER activities align more closely on top of each other in the electrolytes with adjusted pH values (see Figure S17). This experiment therefore supports a non-specific pH effect.

Figure R2. (Figure S17.) OER activity at fixed pH or concentration in freshly prepared (non-purified) alkali hydroxides. (a) CVs in 0.1 M (top) or at a fixed pH of ~13.00 (bottom) (b) CVs in 0.25 M (top) or at a fixed pH of ~13.37 (bottom) (c) CVs in 1 M (top) or at a fixed pH of ~13.76 (bottom). The OER activity at 1.53 V were extracted from CVs for a brief comparison: (d) in 0.1 M (top) and at pH ~13.00 (bottom) (e) in 0.25 M (top) and at ~pH 13.37 (bottom) (f) in 1 M (top) or at pH ~13.76 (bottom. (g) Cross-check of the “cation switching” protocol in the order of KOH – LiOH – KOH. (h) OER activity at 1.53 V plotted against the electrolyte pH (the values in LiOH are omitted). All measurements in this figure were carried out by a “cation switching” protocol to avoid discrepancies in the film thickness. One film was used for different cations; however, a new film was used for each pH-range. The films were rinsed carefully with Milli-Q water before switching. A special pH-electrode was used for these measurements,

which had a ceramic membrane designed to minimize the alkali error (InLab® Routine Pro-ISM, Mettler-Toledo). Despite this, we encountered the same problem in LiOH, and the pH could not be determined. *The reported pH in LiOH are the theoretically predicted values.

Furthermore, since the new pH values in the old purified solutions could not be confirmed with the new pH-meter (since they were already discarded some time ago), we instead present ICP-OES measurements of the old solutions which were performed before they were discarded. The data is shown in Table R1 in is included in the manuscript's Supplementary information as Table S7.

Table R1. (Table S7). ICP-OES of the alkali hydroxides to determine the concentration of the cations (Li⁺, Na⁺, K⁺, and Rb⁺). The concentration of for Cs⁺ could not be determined due to zero emission intensity. The electrolyte pH was calculated from the given concentration according to eqs. (S1)–(S13). It needs to be kept in mind that there is a certain standard error associated with these values, and these should rather be used as an approximation. This is because there is a significant difference in emission intensity and sensitivity between the alkali cations (Li⁺, Na⁺, K⁺, and Rb⁺) at comparable concentrations. Also, since the concentration is measured in ppm, (and not in molarity), it means that the different cations had to be diluted with a different factor. The differences in molecular weight further adds up to this error since the total dilution factor is relatively large (~5000x), and differs between the cations.

	Li	Na	K	Rb	Cs	Ni	Fe
c (M)	c cation from ICP-OES (M)						
0.1 (*purified)	0.061	0.094	0.085	0.079	n/a ^(a)	n/a ^(b)	n/a ^(b)
1 (*purified)	0.61	0.91	0.85	0.80	n/a ^(a)	n/a ^(b)	n/a ^(b)
	Calculated pH						
0.1 (*purified)	12.73	12.95	12.92	12.89	n/a ^(a)	n/a ^(b)	n/a ^(b)
1 (purified)	13.51	13.91	13.84	13.83	n/a ^(a)	n/a ^(b)	n/a ^(b)

^{n/a)} no value obtained or not analyzed, see label for more information.

^{a)} The Cs⁺ ion has zero emission intensity and cannot be determined by ICP-OES.

^{b)} The element could not be detected with ICP-OES, the values are below the detection limit.

Our ICP-OES confirms a similar relative variation in the concentrations between the cations as anticipated from the electrolyte pH. Since the cation trends are easily displaced by differences in the concentration/pH as we showed in Figure R2, mistakes might be crucial for the general cation trend.

We have included the following paragraph on the ICP-OES measurements in the text on p. 8:

Since the pH according to theory increases from small to large cations, it cannot explain the discrepancies in e.g. RbOH (see Table S6). To address this, the concentrations of the cations were determined using ICP-OES, which indeed confirms that the concentration is slightly lower than expected in RbOH (see Table S7). This point towards mistakes during the electrolyte preparation (including uncertainties in the molecular weight due to the hygroscopicity of the salts), rather than an increased amount of e.g. dissolved HCO₃⁻ which could also lower the pH.^{29,56} It should be kept in mind that alkali metal cations are easily ionized elements (EIE), which can introduce uncertainties (see further details in the Methods section).

2) The latter part of the manuscript seems to be more concerned about whether Ni or Fe is the active center in the NiFe oxyhydroxides than the influence of alkaline metal cations. The authors' argument follows the hypothesis that Ni is the active center while Fe is a Lewis acid that enhances Ni activity. Some researchers have challenged this hypothesis by arguing that the Fe is more than a Lewis acid and is actually the active center of the NiFe oxyhydroxides (<https://doi.org/10.3390/molecules23040903>; <https://doi.org/10.1016/j.joule.2018.01.008>). The XAS data and DFT calculation presented in this manuscript do not show further evidence (over previous studies) on Ni being the active center. For example, the XAS data does not prove Ni is oxidized to Ni(IV). Furthermore, despite DFT calculation showed that Fe "edge sites" were the most reactive sites, the authors claimed that the reactivity might too high that they are not catalytically active. But this is purely speculation. I suggest that the authors do not focus on the identification of the active site if they don't have solid evidences. Instead, they should focus on demonstrating that alkaline metal cations have no influence on the NiFe oxyhydroxides, whether Ni or Fe is the active center.

We believe there is a misunderstanding on our arguments on the extension of nickel oxidation, so we will further elaborate on the topic. We did not intent to argue that Ni is the active site, but to state that there is a correlation between the nickel redox peak position and the OER overpotential. It therefore appears as if the oxidized Ni^{3+/4+} (denoted "Ni^{(3+δ)+}" for simplicity) is important for the catalytic turnover rate. However, this does not mean that Ni is active-site and we have pointed this out in manuscript (that now appears in the Discussion on p. 11):

... The lack of a correlation between the alkali cations and Fe-site does not necessarily rule out Fe as the active site, but suggest that the Fe-site is not rate-limiting for the turnover rate...

We do agree on the other hand agree with the reviewer that we could defocus the discussions regarding the active site, and put more stress on the findings regarding the alkali metal cations (which we have now done). We have therefore made changes to the **abstract** as following:

...In situ X-ray absorption spectroscopy (XAS) shows that smaller cations shift both the Ni^{2+/(3+δ)+} redox peak and the OER activity to higher potentials. These modifications can however be explained by a change in the electrolyte pH, which depends on the basicity of the alkali hydroxide. Density functional theory-derived reactivity descriptors support that electrolyte cations impose a negligible effect on the Lewis acidity of Ni, Fe and O lattice sites (and establishes Fe as the strongest Lewis acid), hence supporting a pH effect rather than a "specific" cation effect.

However, just so that the reviewer follows our arguments regarding the correlation between the Ni redox-process and the OER activity. Regardless of Fe being the active site or not, it was shown in the DFT study by Goddard and co-workers, that high-valent Fe(IV) provide efficient formation of an O radical, whereas the Ni sites actually is more probable to catalyse O-O bond coupling, thus pointing to a bimetallic "synergy" between the two sites (PNAS, 2018, 10.1073/pnas.1722034115). In line with that, the mentioned review article by Hunter et al. (Molecules, 2018, doi.org/10.3390/molecules23040903) also argued that Ni(IV) oxo complexes are less stable than high-valent Fe-oxo complexes. A "synergy" is in fact supported by the DFT study by Friebel et al. (JACS. 137, 2015, doi 10.1021/ja511559d), where the OER intermediates at the Fe-site have "optimal" adsorption energies only in the bimetallic Ni-Fe(OOH) catalyst. This suggests that the neither the Ni- or Fe-sites in the respective monometallic catalysts are identical to the corresponding sites in the Ni-Fe(OOH) catalyst, and their catalytic abilities/reactivities cannot be directly compared. In fact, the experimental arguments for oxidized Ni^{3+/4+} being important for the catalytic rate has been shown by Boettcher and co-workers (JACS, 2014, doi 10.1021/ja502379c), which is correlated to the oxidized NiOOH lattice being required for a high through-film conductivity (since reduced Ni²⁺(OH)₂ is an

insulator). This could in fact be the reason why we observe the correlation with the oxidation of $\text{Ni}^{2+} \rightarrow \text{Ni}^{3+/4+}$. It became more obvious during the review that this correlation is more complex, which will be more thoroughly discussed in the next comment.

The exact mechanism of how the Fe-site catalyses OER is a matter of controversy we believe that this is beyond the scope of this study, which also the reviewer pointed out (and we agree). We have tried to be more careful in the discussions regarding the active site, and we also propose a change in the title to following;

“Key activity descriptors and catalytic sites in Ni-Fe oxygen evolution electrocatalysts in the presence of alkali metal cations”

Then, it is true that we cannot for sure identify the changes at the Fe-site as an increase in the oxidation state to Fe^{4+} , however, both the Ni *K*-edge position (~ 8346 eV) and the shortening of the Ni-O coordination distance (~ 1.88 Å) beyond doubts claim an overall oxidation state of $\sim \text{Ni}^{3.8}$ at OER potentials (the absolute oxidation state depends on the cation and the potential, and therefore we denote this state as “ $\text{Ni}^{(3+\delta)+}$ ”, which should not be confused with a state that does not contain Ni^{4+}). At 1.66 V, we find that ~ 100 % of the Ni-centers are oxidized to Ni^{3+} , and of these a fraction of ~ 80 % are further oxidized to Ni^{4+} , which also agrees with other studies of similar catalysts. We are aware that sometimes lattice-incorporated Fe prevent the detection of the high-valent $\text{Ni}^{3+/4+}$, but this should also be visible as a smaller *K*-edge shift (which is not the case herein), and seems to vary for different Ni-Fe catalysts depending on experimental and unknown conditions (Görlin, JACS, 2017, 10.1021/jacs.6b12250).

We have inserted two paragraphs on p. 6 to clarify the changes associated with the Ni *K*-edge and the Ni-O bond distances:

At higher potentials (1.66 V), the Ni *K*-edge is consistent with a complete oxidation from $\text{Ni}^{2+} \rightarrow \text{Ni}^{3+}$, and an additional fraction of ~ 80 % Ni^{4+} (see Figure 3e and Table S3).

The Ni-O bonds agree with the Ni *K*-edges, showing oxidation of $\text{Ni}^{2+} \rightarrow \text{Ni}^{3+}$, and additionally ~ 70 % of Ni^{4+} (see Figure S9c).

Then, we agree that the changes at the Fe-site are indeed more difficult to interpret due to the absence of a strong Fe *K*-edge shift, and that this should have been discussed in more detail in the manuscript. The shortening in the Fe-O bond distances (~ 1.94 Å) would support ~ 90 % Fe^{4+} , however the edge position does not support oxidation at all (0 % Fe^{4+}). This discrepancy was discussed in the review article by Hunter et al. (Molecules, 2018, doi.org/10.3390/molecules23040903) mentioned by the reviewer, which is now cited in the manuscript. In that article they discuss the possibility of a spin-state transition from high-spin Fe(III) to low-spin Fe(III), which we find interesting. We did not consider this at first because the low-spin state is very unusual for iron oxides at NTP conditions. However, it might be that the geometric “strain” effect that an oxidized NiOOH lattice possibly imposes on the Fe-site simulates elevated external pressures, which might then force a spin-state transition. (We have included a comment on this in the XAS section.) In the other mentioned article by Hunter et al. (Joule, 2018, doi.org/10.1016/j.joule.2018.01.008), high-valent Fe(+4,+5,+6) species were observed in acetonitrile-based studies. These high-valent species were found to be minor (~ 3 %), where the major phase was also confirmed to be high-spin Fe(III) state according to Mössbauer. The high-valent Fe species did on the other hand react with the aqueous hydroxide ion, where the spectral signal was quenched after addition of KOH. This might explain why the high-valent Fe species are rarely observed during similar conditions as in this study (i.e. aqueous electrolyte). What we find still puzzling, is that we observe that as much as ~ 90 % of the Fe-centers that undergoes Fe-O bond

contraction and possibly enters the Fe⁴⁺-state, whereas only ~3 % was observed to enter the high-valent Fe state in the study by Hunter et al. (i.e. mostly at edge or edge or corner sites). We have made sure to include a paragraph on this in the XAS section. Please note that we have moved the XAS section up, it now appears directly after the electrochemical section and before the pH-series. We think that the outline improved this way.

We have added the following paragraph on this in the Discussion section on p.11:

Michael et al.³⁰, observed using Raman spectroscopy that CsOH promotes a longer Ni-O bond in the Ni(Fe)OOH catalysts compared to LiOH. Our XAS data instead supports a shorter Ni-O bond, which just reflects the higher oxidation state in CsOH. The changes related to Fe XAS spectra is a matter of controversy, and we do not find a particularly strong correlation with the electrolyte cation. The lack of a correlation with the Fe-site does not necessarily rule out Fe as the active site, but suggest that the Fe-site is not rate-limiting for the turnover rate. The Fe-O bond is consistent with ~90 % of the centers being oxidized to Fe⁴⁺, however, Dau and co-workers did not regard this to an increase in the oxidation state due to the lack of an edge shift.^{13,15} According to other studies, a small fraction (3- 21 %) of high-valent Fe-species (+4 and beyond) are formed.^{19,63} Another possibility mentioned by Hunter et al.,⁶⁴ is a spin-state transition from high-spin Fe³⁺ to low-spin Fe³⁺, which would cause a similar spectral change as both charge transfer and geometry effects and consistent with the changes in our spectra.⁶⁵ This will have to be addressed in future studies.

3) The data in the manuscript may need to be analyzed with more care. For example, it is claimed that there is a strong correlation between the Ni²⁺/Ni³⁺ redox peak potential and the OER potential because there is a constant separation between the two processes (Figure 4) and both exhibit near-identical slopes vs pH (Figure 5c). However, when looking carefully into the data in Figure 4, the potential separations are not as “constant” as the authors claimed. The slope fitting for OER in Figure 5c also seems not to include all the data points. Since direct conclusions are drawn from these analyses, it should be taken with great caution. There is also a previous study by Boettcher’s group which demonstrated that the Ni²⁺/Ni³⁺ redox peak potential did not correlate with water oxidation activity (<https://doi.org/10.1021/acscatal.5b02924>).

We thank the reviewer for this suggestion, we have now gone through all data again and discovered that the picture is a bit more complex than we initially thought. We actually see that the correlation is not that strong to the absolute levels of oxidized Ni^{3+/4+} during steady-state at higher potentials (i.e. after reaching complete oxidation), but the correlation is stronger to the onset potential of the oxidation (i.e. the peak-peak position). This might explain some of the discrepancies we observe in the in situ XAS data between the NaOH and CsOH cations as well, since the differences in the overall K-edge shifts are fairly small (we will discuss this in more detail in next Comment #4).

In the Discussion section of the manuscript on p. 11 we have modified the text as following:

Our XAS and electrochemical data both support that the electrolyte pH affects the Ni^{2+/(3+6)+} redox process and the OER activity in a similar fashion, where large cations shift both the oxidation and the OER process to lower potentials. In recent findings by Pasquini et al.⁵⁸ the catalytic rate of a CoO_x catalyst was found to be controlled by the levels of oxidized Co⁴⁺. A similar scenario could prevail in our Ni-Fe catalyst, although we do not find as strong correlation with the population of oxidized Ni⁽³⁺⁶⁾ during steady-state conditions. The correlation might also be related to the importance of oxidized Ni^{3+/4+} due to the associated increase in the through-film conductivity, as was shown by Boettcher and co-workers.

Then, the fit lines in Figure 5c in the previous version of the manuscript were indeed performed without taking all the data points into account, as the reviewer observed. We have changed this now, and all data points are included in the fit. However, what became more evident, is that the absolute values of the pH-slopes for both the $\text{Ni}^{2+/(3+\delta)+}$ redox peak and the OER overpotential are dependent on the experimental conditions such as the scan-rate, concentration, and the current density. Both slower scan-rates and lower current densities decrease the pH-slopes, closer to steady-state conditions, the slopes are more like $\sim -40 \text{ mV pH}^{-1}$ instead of $\sim -80 \text{ mV pH}^{-1}$. The processes are still non-Nernstian, however, this is typical for mass transfer, since the peak separation between the anodic ($E_{p,a}$) and cathodic ($E_{p,c}$) redox peaks also increases at higher scan-rates and at lower concentration. We have therefore extracted more data shown in Figure R3 and R4 in this letter (and as Figure S15 and Figure S18 in the Supplementary Information). Then, what is more striking (and that we missed out before) is that the cathodic redox peak is much less affected by the pH than the anodic peak, and exhibit more or less a Nernstian pH-slope ($\sim 0 \text{ mV pH}^{-1}$ in the RHE scale; or it varies between -8 and $+26 \text{ mV pH}^{-1}$ depending on the scan-rate, see Figure R4b-c). We cannot fully explain this effect, however, it might indicate that different species are involved in the metal oxidation and the reduction steps, but is beyond the scope of this study.

We have modified the text slightly on p. 8 in the manuscript as following:

Both the peak positions ($E_{p,a}$) and the OER activity respond in the same direction of fairly similar magnitudes ($\sim 40\text{-}50 \text{ mV}$), however, the separation between the two processes varies between $\sim 70\text{-}95 \text{ mV}$. In Figure 4c, it is visible that OER activity increases with the concentration, and the differences between the cations become more pronounced (see also Figure S15a-c). Smaller cations however show a stronger saturation at higher concentrations than large cations. A slight change in the activity trend is also visible, where the “dip” in RbOH is less pronounced at higher concentrations (Figure S15 d-e).

We have also added following a paragraph on p. 8 in the manuscript where we clarify the dependence on the experimental conditions:

The $\text{Ni}^{2+/(3+\delta)+}$ redox-peak and the OER overpotential both exhibit non-Nernstian pH-slopes of ca. -80 mV pH^{-1} in the RHE scale (Figure 4e). The absolute pH-slopes are dependent on the experimental conditions, and decrease to ca. -40 mV pH^{-1} closer to steady-state conditions (see Figure S18a-c). This is typical when mass transfer limitations are involved, as also inferred by the response of the redox-peak to both the scan-rate and pH (see Figure S18d,g). Since the uncompensated solution resistance (R_u , iR-drop) drastically increases at low concentrations, it might as well affect the kinetics (see Table S8). The cathodic redox peak ($E_{p,c}$), is, surprisingly, less affected by the electrolyte pH than the anodic peak ($E_{p,a}$), and actually has a close to Nernstian pH-slope (ca. 0 mV pH^{-1} in the RHE scale), which we will come back to in later discussions (see Figure S18b-c). The absolute level of redox charge (Q_{peak}) is not that greatly affected by the concentration as the peak potential as inferred in our data above, however, both higher pH and slower scan-rates help approach the unity value of $\sim 1.8 \text{ e}^-$ per Ni^{-1} (see Figures S18e-i).

We have also added a paragraph to the Discussion on p.11 on the interpretation:

Both the OER activity and anodic redox-peak exhibit pH-slopes that deviates from Nernstian (ca. -40 to -90 mV pH^{-1} in the RHE).⁵⁹ In agreement with earlier studies,^{27,48} this has been attributed to non-concerted proton-electron transfer steps, in accord with ~ 2 protons transferred per 1 electron (H^+/e^- ratio of ~ 2). Soft XAS studies at the O K-edge have also reported hole states on the oxygen ligands related to O^{1-} as a result of charge transfer.^{23,24,60–62} In work by Koper and co-workers,^{25,26} a Raman band around $\sim 800\text{-}1150 \text{ cm}^{-1}$ has revealed similar deprotonation pathways of adsorbed

intermediates (OH_{ads} or OOH_{ads}) leading to peroxo-like intermediates (O_{ads}^- or $\text{O}_{2,\text{ads}}^-$) denoted “active oxygen”. Garcia et al.³⁶ found a higher intensity of this Raman band in CsOH compared to LiOH. This band was in earlier studies from their group shown to intensify at higher electrolyte pH.^{25,26} Since larger cations (i.e. CsOH) results in a higher electrolyte pH, it could explain why larger cations promote this band. The nature of the reactive species could in turn explain why a higher pH promotes a higher OER activity.

Figure R3. (Figure S15). The influence of alkali hydroxide concentration on the OER activity. Measurements of the $\text{Ni}_{65}\text{Fe}_{35}$ catalyst in purified 0.05 M, 0.1 M, 0.25 M, 0.5 M and 1 M hydroxides (CsOH, RbOH, KOH, NaOH, and CsOH). (a) CVs at 10 mV/s. (b) TOF_{mass} at 1.53 V vs. the concentration. (c) TOF_{mass} at 1.56 V vs. the concentration. (d) Current density (j) and TOF_{mass} at 1.53 V in 0.1 M XOH. (e) j and TOF_{mass} at 1.53 V in 1 M XOH. (f) Experimental pH of the alkali hydroxides measured with a standard pH-meter (the pH of LiOH is exclude from the plot due to erroneous reading). (g) Theoretical pH of the alkali hydroxides calculated from their pK_{b} values using eqs. (S1)-(S13). (h) j at 1.48 V vs. the electrolyte pH. (i) j at 1.53 V vs. the pH. (j) j at 1.56 V vs. the pH. (k) TOF_{mass} at 1.48 V vs. the pH. (l) TOF_{mass} at 1.53 V vs. the pH. (m) TOF_{mass} at 1.56 V vs. the pH. The pH of LiOH in figures (h)-(m) were determined from the calibration curve in Figure S16.

Figure R4. (Figure S18) Influence of the electrolyte pH on the peak and OER potential of the Ni₆₅Fe₃₅ catalyst. (a) The OER overpotential extracted at two different current densities (1 mA cm⁻² and 10 mA cm⁻²), plotted vs. the pH. (b) Anodic ($E_{p,a}$) and cathodic ($E_{p,c}$) redox peak potentials of the Ni²⁺/^{(3+δ)+} couple determined from CVs at 2 mV/s plotted vs. the experimental pH. (c) Peak potential from CVs at 10 mV/s vs. vs. the experimental pH. (d) Peak potential from CVs at 10 mV/s vs. the concentration. (e) The integrated redox peak area (charge, Q_{peak}) from CVs at 10 mV/s in units 'moles of electrons transferred per moles of Ni ions' (e⁻ per Ni), plotted vs. the concentration. (f) Q_{peak} from CVs at 10 mV/s vs. the electrolyte pH. (g) Peak potential from CVs at 2 mV/s vs. the concentration. (h) Q_{peak} from CVs at 2 mV/s vs. the concentration. (i) Q_{peak} from CVs at 2 mV/s vs. the electrolyte pH. Note that the pH of LiOH in all the plots in this figure is determined from the offset between the RHE and the Ag/AgCl electrodes.

We have also merged old Figure 4 and 5 into the new Figure 4 (shown as Figure R5 here in this rebuttal letter), where we exchanged the data to conditions where the pH-slopes are more similar. The rest of the data are shown in Figure S15 and S18 and discussed in the text.

Figure R5. (Figure 4). The activity of Ni₆₅Fe₃₅ catalyst in selected alkali metal cation hydroxides with different molar concentrations (0.05 M, 0.25 M, 0.25M, 0.5 M, and 1 M in CsOH) and pH. (a) CVs at 10 mV/s to demonstrate the peak shift induced by higher concentrations. (b) CVs at 10 mV/s to demonstrate the typical peak shift induced from small to large cations. (b) Current density (*j*) at 1.53 V vs. RHE plotted against the electrolyte concentration (c) TOF_{mass} at 1.53 V vs. RHE plotted against the experimental electrolyte pH. (d) The OER potential at 5 mA cm⁻² (filled circles) and the anodic peak potential (*E*_{p,a}) of the Ni²⁺ ↔ Ni^{(3+δ)+} redox peak extracted from CVs at 10 mV/s (up-pointing arrows). The color code for the alkali metal hydroxides in the legend in (c) applies also to (d)-(e). The pH in LiOH was determined with an alternative method, and is shown in (d), however, is excluded from the fit line in (d).

Now back to the last concern regarding the findings of Boettcher and co-workers, where they found a lack of correlation between the Ni^{2+/3+} redox peak and the OER activity upon intentional insertion (or doping) of different transition d-metal elements (<https://doi.org/10.1021/acscatal.5b02924>). The elements (or cations) in their study all have d electrons in the outer valence shell (Ti 3d⁴, Mn 3d⁷, La 5d¹, Ce 5d¹), whereas the alkali metal cations in this study only have s-electrons; Li 2s¹, Na 3s¹, K 4s¹, Rb 5s¹, Cs 6s¹. The metal s-cations in this study (especially K⁺ and Na⁺) are basically always present in OER studies in alkaline electrolytes (they also used 1 M KOH in the mentioned study) and according to our SEM-EDS analysis post OER (Figure S3 and Table S2) neither of these 1s-block cations “specifically” incorporate to replace lattice sites in the Ni-Fe(OOH) films. The answer to why the Ni redox-peak in the article from Boettcher and co-workers did not correlate with the OER activity, is most likely that the incorporation of the d-elements into the Ni(OOH) lattice sites occur in their study (i.e. denoted Ni_{1-z}M_zO_xH_y, where z = Mn, Ce, Fe, La, Ti), which will impose an “electronic” effect and thereby shift the Ni^{2+/(3+δ)+} redox peak rather unpredictably. This “electronic” effect can be described as a change in the d-electron count of the metal oxide, and other specific effects between the two neighbouring sites such as an impact on the metal-oxygen (M-O) covalency and the Lewis acidity (as we show upon Fe-incorporation in the NiOOH lattice using DFT in this study). The metal d-electrons (and the e_g-filling level) are both known to be correlated to the OER activity, as demonstrated by

Shao-Horn and co-workers for perovskite oxides (Suntivich, J. *et al. J. Phys. Chem. C*, 2014, doi 10.1021/jp410644j). As Boettcher and co-workers show in several other studies, peak potentials between different bimetallic and monometallic catalysts (e.g. Co-Fe and Ni-Fe, Co, Mn, Ni) are distinct and there is no simple correlation between redox-peak potential and the OER activity to the best of our knowledge for transition metal oxides. So, what the effect d-metal incorporation has on both the redox-peak and the OER activity of the Ni(OOH) catalyst is rather unpredictable by experiment. Furthermore, they did not employ an in situ technique that can specifically assign the redox-peaks to a specific metal site, so they in principle do not know whether the visible redox-peaks correspond solely to $\text{Ni}^{2+} \rightarrow \text{Ni}^{3+/4+}$ oxidation. Also, they did not investigate how the electrolyte pH influences the redox-peak for each of their d-element doped $\text{Ni}_{1-z}\text{M}_z\text{O}_x\text{H}_y$ catalysts, so we cannot draw further conclusions regarding the lack of correlation between the OER activity and the nickel redox peak. We hope that this is enough to convince the reviewer that the correlation between the Ni redox peak and the OER activity for d-metal incorporation is different than the effect of having s-block alkali cations in solution.

4) The “exceptions” observed in the data should also be considered more carefully. In analyzing the Ni K-edge shifts in different electrolytes, the shift increases in the order similar to the activity trend but with an “exception”. However, one might also argue with this “exception” that Ni K-edge shift is not a real indicator of OER activity.

We agree that this exception in the in situ XAS might indicate a lack of correlation with the OER activity. However, we have investigated this in more detail:

First, we have evaluated the standard error from repetitions of different films at the occasion of the beamtime, and we have also compared two sets of data from two different beamlines. The data set we currently show in the manuscript was from the P64 beamline at Petra III (DESY, Hamburg). The other data set that we recorded a couple of months earlier is from the SuperXAS-X10DA beamline (SLS, PSI, Switzerland). We did not repeat all the hydroxide solutions at each of the beamtimes (there was not enough of time), however, selected hydroxides were repeated two times. Comparing the two data sets allow us to obtain some information regarding the overall standard error for the cation trends (see Figure R6).

Figure R6 is now included in the manuscript’s Supplementary Information as Figure S11.

The reason why we did not show both data sets in the manuscript before was because there are significant differences between the two experimental conditions. We also had troubles with the electrochemistry during the SuperXAS - SLS beamtime (most likely due to counter electrode problems), and we did therefore not have enough of time to collect the Fe K-edges. The major differences between the two data sets from the respective Petra III and SLS beamtimes are following; **(1)** slightly different Ni:Fe compositions ($\text{Ni}_{65}\text{Fe}_{35}$ vs. $\text{Ni}_{75}\text{Fe}_{25}$) **(2)** different film thickness (300 nm vs. 50 nm) **(3)** different applied electrode potentials during the K-edge collection **(4)** different measurements protocols (at Petra III we always used a new catalyst film for each new cation and at SLS we re-used the same catalyst film for several different cations).

Because of the smaller metal loading and film thickness at SLS, the OER activity was significant lower (i.e. higher overpotential/onset), and therefore both the activity and redox-peaks are shifted to higher potentials for the Ni₇₅Fe₂₅ catalyst at SLS. Therefore, the comparison of the K-edge shift

Figure R6. (Figure S11). Comparison of the Ni K-edge positions from two different beamlines (P64, Petra III, DESY) and SuperXAS-S10DA, SLS, PSI) (a) Ni K-edges of the Ni₆₅Fe₃₅ catalyst (different samples measured at the same potential, 1.48 V vs. RHE) at P64-Petra III, which is the main data presented in this manuscript. This catalyst had a metal loading of $\sim 25 \pm 2 \mu\text{g cm}^{-2}$, corresponding to a film thickness of $\sim 300 \text{ nm}$. (b) Ni K-edges of a Ni₇₅Fe₂₅ catalyst (different samples at the same potential, 1.54 V) at the SuperXAS-SLS. This catalyst had a metal loading of $3 \pm 1 \mu\text{g cm}^{-2}$, corresponding to a film thickness of $\sim 50 \text{ nm}$. (c) The Ni K-edges in different alkali hydroxides measured at P64-Petra III. (d) The Ni K-edges in different alkali hydroxides measured at SuperXAS-SLS. (e) Trend plot of the Ni K-edge positions from P64-Petra III. (f) Trend plot of the Ni K-edge positions from SuperXAS-SLS. The potentials shown here are selected based on the K-edge shifts, which are supposed to represent a state right after the onset of the metal oxidation, so before the highest oxidized state is reached. These potentials were selected for comparison since the absolute redox-peak positions and the OER activities differs between the two catalysts because of different metal loadings ($\sim 300 \text{ nm}$ at P64-Petra III versus $\sim 50 \text{ nm}$ at Super XAS-SLS). Note that new catalyst films were always used for each measurement at Petra III, whereas one film was used for several different electrolyte cations at SLS via a “switching protocol”. The films were rinsed with plenty of Milli-Q water in between the switching.

had to be done at different potentials where the shifts are comparable (1.48 V at Petra III and 1.54 V at SLS). Despite these differences, the overall “cation trend” is well reproducible between the two data sets, increasing from small to large cations with a “dip” in RbOH. The exception between CsOH and NaOH is not observed in the data set from SuperXAS-SLS. Also, comparing edges from different films shows that there is a certain standard error associated with the edge shift. The differences in the overall trends from the XAS spectra at Petra III might therefore also be influenced by small variations (uncertainties) in the film thickness (since we used new films for each new hydroxide/cation). There are also other factors that might contribute to discrepancies in the OER activity; the CVs that were always (and only) measured at the end of the XAS before changing catalyst film, and these could have suffered from pile-up of O₂ bubbles as was discussed in Comment #1 above. So, the XAS measurements are indeed prone to experimental error introduced from variations in film thickness and O₂ bubbles.

We further stress that the differences between the *K*-edge positions across the cation series in the in situ XAS measurements are very small in overall (the differences in electrolyte pH is also not that large in 0.1 M solutions), and therefore the most pronounced difference is to expect between the two extremes (LiOH and CsOH), where also the difference in the pH is largest. Also, since the correlation between the OER activity appears to be stronger with the Ni^{2+/(3+δ)+} peak position rather than to the number of oxidized Ni^{(3+δ)+} centers in the “equilibrium state”, hence the largest difference is to expect close to the redox-onset, so before reaching the equilibrium levels of oxidized Ni^{(3+δ)+}.

We have added the following text to the XAS section on p. 7 in the manuscript:

Noteworthy is that the differences are most pronounced at quasi-equilibrium (in the “OX” region) and diminish at higher potentials (in the “OER region”), meaning that the cations do not significantly alter the equilibrium levels of oxidized Ni^{(3+δ)+}. Repetitions and a cross-check of a data set from a different beamline (SuperXAS-XD10 at SLS, PSI) shows that the differences in the Ni *K*-edge positions between the cations are relatively small, and in fact close to the overall standard error (see Figure S11). The edge shift and the OER overpotential are also influenced by the film thickness, whereby variations in catalyst loading might introduce further discrepancies. From an overall perspective, the cation trend is well reproducible, where smaller cations shift the oxidation and the OER processes to higher potentials compared to larger cations.

5) Regarding the iR correction, was the resistance (R) of every sample measured independently? If so, the experimental details for determine the resistances should be provided. If not, it is highly recommended to do so since they may be different in different electrolytes.

Indeed, automatic iR-compensation was always applied independently for each catalyst film. The uncompensated solution resistance (R_u, iR-drop) was applied to all potentiostatic techniques to a level of 85 % in the software of the Bio-Logic potentiostat (EC-Lab, the “ZIR” technique). A higher level than 85 % could not be applied, especially at higher electrolyte concentrations where clear overcompensation effects were observed (demonstrated in Figure R7). This phenomenon is well-known, and therefore the automatic iR-compensation level should not be too high to avoid this, however, a high level as possible is desirable. To compensate for the remaining 15 % is also a matter of discussion. We of course tried out different levels of “extra” iR-compensation to reach as close to 100 % as possible, however, applying more than a total level of 90 % also generated overcompensation effects, as it is depicted in Figure R6. We therefore chose to compensate for a total level of 90 % for all measurements. We have gone through the experimental data again, to make

sure that all measurements have a total level of 90 % iR-compensation. This did not change much in the relative trend across the cations as we could see, however, some of the absolute values changed slightly.

Figure R7. Cyclic voltammograms (CVs) of the Ni-Fe catalyst with different level of automatic iR-compensation in (a) 0.5 M KOH and (b) 1 M KOH. The automatic level was set to 80 % for in the potentiostat, and the additional levels were carried out manually afterwards. A total level of 90 % was applied to avoid overcompensation effect sometimes observed between 95-100 % iR-compensation levels. *This figure is not shown in the main manuscript.*

To make clear our methodology of iR compensation, we have now added the following paragraph to the Methods section (p.13):

The uncompensated solution resistance (R_u , iR-drop) was determined individually using electrochemical impedance spectroscopy (EIS) at 10 kHz, and automatic iR-compensation was applied to a level of 85 % (using the ZIR method in the EC-Lab software). An additional 5 % iR-compensation was applied afterwards (total level of 90 %) to avoid overcompensation. A typical value of the iR-drop in the in situ XAS setup was 25 Ohm in 0.1 M XOH solutions.

6) Regarding the unexpected drop in activity in RbOH, have the authors tried a different batch of RbOH since they suspected it might have larger carbonates or water contaminations?

In fact, we did try different batches of all the alkali metal hydroxide salts (identical to the old ones according to the supplier). These were the measurements now shown in Figure R2 in this rebuttal letter (and attached as new Figure S17 in the manuscript's Supporting Information). In the new hydroxide batches, the "drop" in RbOH is slightly less pronounced, and we mostly observe this "dip" in 0.25 M. We point out that this "dip" in the old purified hydroxide solutions were mostly pronounced at low concentrations (i.e. 0.05 M to 0.25 M), and at higher concentrations (0.5 M and 1 M), this dip was not seen in RbOH. From the pH-values presented in Table S6 in the Supporting information, it is also evident that the OER activity in RbOH more or less does follows the trend in the electrolyte pH. We do not have the old batches of electrolytes saved (they were discarded some months ago now after we finished the measurements). As we mentioned in Comment #1 above (shown in Table R1), this dip in the old and purified RbOH solution could as well be confirmed using ICP-OES as a dip in the molar concentration. This suggests that the OER activity indeed follows the relative differences in the electrolyte pH.

7) Some other minor issues:

Page 1: metal-air batteries are not relevant to the discussion of H₂ production from water.

We agree, that since we did not investigate both OER and ORR the relevance is smaller for metal-air batteries, however, we kept regenerative fuel-cells since OER is the heart of electrolysis. So, we have removed the metal-air batteries from the sentence.

Page 3: the TOF value in LiOH should be 0.04.

Fixed.

Page 4: the compositional inhomogeneity shown in Figure S3 and Table S2 is not “small”.

We have changed the text on p. 5 to as it follows:

The Ni-Fe catalyst films exhibit a significant inhomogeneity in the Ni:Fe composition, where several local areas have a clear Fe-enrichment (see Figure S3c-d and Table S2).

Page 5: Figure S5b should be Figure 5b.

We thank the reviewer for point out some typos in the labels and in the caption text, they have been fixed.

Author's Response to Reviewer #2:

Comments and Suggestions for Authors of: NCOMMS-20-10153

General comments:

The authors present a study of key activity descriptors for the OER reaction on Ni-Fe oxyhydroxide catalysts and conclude that pH is the dominating one, rather than a specific cation effect. The manuscript is largely well written, and I agree with the authors conclusions. For the field of electrolysis, it is important to clarify the role of different descriptors. However, there are a number of points, especially in the practical work where the authors need to specify, explain, and motivate better. On a few points it might also require some additional analysis or measurements. For many of the parameters presented, the difference between the different cations are within or close to the error margin, thus it becomes very important to show that the authors really have done the practical work and the analysis correctly.

We thank the reviewer for the valuable feedback on our manuscript.

Major Issues

- 1. There must be a typo in the numbers regarding results of TOF where it says: "The intrinsic activity was evaluated as TOF_{mass} (turnover frequency per total metal ion loading), which justified the geometric activity trends (Figure 2c), with variations between 0.4 s⁻¹ (LiOH) to 0.1 s⁻¹ (CsOH) (see Table S1)". From fig 2c and table S1 it should be 0.04 s⁻¹?**

First, indeed, it was a typo in the TOF_{mass} given in the text (it should have been 0.04 s⁻¹ and not 0.4 s⁻¹ in LiOH). Second, when we double checked all the TOF calculations, we found an error in the calculation for the 0.1 M alkali metal hydroxides that was presented in Table R2 including these values. We had forgot to divide the values by z (4), so the values should have been ~4 times lower. We did not make the same mistake for the other TOF_{mass} values for the pH-series presented in old Figure S5 (new Figure S15) where the values indeed were ~4 times lower. It is normal that TOF is relatively low in this case, because the total metal loading is relatively high (~25 μg cm⁻², ~300 nm film thickness) and the concentration is low (0.1 M), so the lower values make more sense when comparing to literature and from our own experiences (see the review article by Dionigi et al., 2016, doi.org/10.1002/aenm.201600621). It was also shown by Görlin et al. that TOF_{mass} increases with lower loading and with higher electrolyte concentration (doi.org/10.1021/jacs.6b12250).

Since the mistake was a constant, the relative cation trend did not change after this correction in TOF_{mass}. We have added the new values in the main text of the manuscript, and updated Table S1 with the information in Table R2. For clarity, we have also added all the other OER activity parameters for the complete concentration (pH) series in the new Table S8 in the supplementary information. After other changes including moving the in situ XAS section up, some of the data in the new Figure 2 was exchanged for the OER overpotential at 10 mA cm⁻² and the Tafel slopes.

We have also inserted following text in the main manuscript on p. 4:

This might appear low at first glance, however, agrees with reports of other Ni-Fe catalysts at similar conditions.⁶ The turnover frequency is also known to depend on the experimental conditions, and is often lower for thicker films and in lower molar concentrations.^{48,49}

Table R2. OER activity parameters of the electrodeposited Ni₆₅Fe₃₅ catalyst measured in purified 0.1 M alkali hydroxides (LiOH, NaOH, KOH, RbOH, CsOH). The TOF values were obtained from the data collected from the potential steps at the occasion of the in situ XAS measurements (these were measured using a leak-free Ag/AgCl

reference electrode), whereas the overpotential (η_{OER}) at 10 mA cm⁻² and Tafel slopes were measured at a different occasion “in house” using a reversible hydrogen electrode (RHE) as reference. All values are reported on the RHE scale.

Electrolyte (0.1 M)	Cation radius (pm) ^(a)	iR-drop (Ω)	Loading (ICP) ($\mu\text{g cm}^{-2}$) ^(b)	j @ 1.48V (mA cm^{-2})	TOF @1.48 V (s^{-1})	j @ 1.56 V (mA cm^{-2})	TOF @ 1.56 V (s^{-1})	η_{OER} @ 10 mA cm ⁻² (mV)	Tafel slope (mV dec^{-1})
LiOH	69	36 (13)	21 (4)	0.19 (0.08)	0.009 (0.003)	2.0 (0.4)	0.009 (0.003)	352 (20)	41 (5)
NaOH	102	31 (12)	27 (1)	1.03 (0.21)	0.016 (0.003)	4.3 (0.1)	0.016 (0.003)	309 (2)	37 (1)
KOH	138	31 (11)	26 (1)	1.03 (0.09)	0.015 (0.004)	4.0 (0.7)	0.015 (0.004)	316 (17)	38 (1)
RbOH	149	31 (16)	23 (2)	0.87 (0.19)	0.014 (0.004)	3.1 (0.6)	0.014 (0.004)	331 (27)	39 (1)
CsOH	170	33 (10)	21 (2)	0.99 (0.23)	0.023 (0.003)	4.9 (0.4)	0.023 (0.003)	295 (5)	38 (1)

^(a)The cation radius was obtained from Marcus et al.⁸
^(b)The Ni:Fe compositions and loadings were determined after the electrochemical characterization using ICP-OES.

2. I am missing information and discussion on the purity of the different electrolytes. The authors present a rigid method employed for electrolyte purification in the SI, but I don't find information on the purity of the starting chemicals. It would also be good to measure or estimate the types and concentrations of impurities in the electrolytes used. As the differences between the different solutions are so small, and this type electrochemical reactions are highly sensitive to contaminations, this information is of high importance.

We have added the purity grades of the different alkali hydroxides to the Methods section in the main manuscript on p. 13:

The alkali metal hydroxides used for activity investigations were following; LiOH (monohydrate, 99.995 % trace metals basis, Sigma-Aldrich), NaOH (monohydrate $\geq 99.996\%$ metals basis, Alfa-Aesar), KOH (semiconductor grade, pellets, 99.99%, Sigma-Aldrich), RbOH (hydrate, Sigma-Aldrich), CsOH (monohydrate, 99.95% trace metals basis, Sigma-Aldrich).

We have also made efforts to analyze the electrolytes for impurities, both before and after the purification step. We used ICP-OES to analyze for selected trace impurities (on the ppm level). We screened for the following elements; Li, Na, K, Rb, Cs, Ni, Fe, Mn, Co, Cu, Zn, Ca. Please note that Cs⁺ could not be detected (it has relatively low emission intensity). We did indeed find trace impurities of ~ 5 ppm Fe and ~ 3 ppm Zn in RbOH, and small amounts of Fe (less than $\sim 1\%$) in CsOH. Both these impurities were removed after the purification step, and we could not find any other impurities in the purified alkali hydroxides. Since we inserted an additional filtering step in the last step of the purification procedure, this might have removed otherwise possible impurities of Ni commonly introduced during the purification step. We also checked that there were no other major impurities using SEM-EDS elemental mapping, which we acquired using a Zeiss LEO1550 microscope. For this analysis, 1M alkali hydroxides were drop-casted on H₂SO₄-cleaned glassy carbon plates (4 x 4 mm, SIGRDUR® K, HTW), and dried overnight. Elemental maps were collected of larger areas (0.3 x 0.2 mm), for 10-15 min, and 2-3 spots were investigated for each salt. The acceleration voltage was 15 kV. Clear signals of the alkali metals were visible in the EDS spectra except for Li which is a too light element to be detected in EDS. No other impurities were visible except for carbon and oxygen. The empty glassy carbon plate (control sample) contained 97 at. % C and 3 at. % O.

Figure R8. (Figure S4). ICP-OES of the alkali hydroxide electrolytes addressing impurities. Only the relevant spectra are shown. All electrolytes had been diluted to a concentration of ~ 1000 ppm with respect to the cation (not shown here). (a) as-received alkali hydroxides (b) purified alkali hydroxides. As can be seen, the as-received RbOH contains ~ 5 ppm Fe and ~ 3 ppm Zn impurities. In CsOH there are only slightly elevated levels of Fe visible, estimated to ≤ 1 ppm considering the noise level. (c) EDS spectra recorded of the 1 M purified hydroxides dried on H₂SO₄ cleaned glassy carbon plates (4 x 4 mm, SIGRADUR® K, HTW) using the build-in detector in the Zeiss LEO1550 microscope to cross-check for other impurities. Elemental maps were recorded for ~ 10 -15 min at 2-3 different areas (300 x 200 μm spot size) for each electrode, using an acceleration voltage of 15 kV. The variations in the carbon content in the EDS spectra depends on how homogenous the distributions of the dried salts are, and has no scientific significance. We do also not report the O-content since the accuracy is low for soft elements, but was on average 66 ± 9 at. % for the hydroxides, and ~ 3 at. % for the empty glassy carbon plate. It should be noted that Li is a too light element to be detected in EDS, and Cs could not be detected using ICP-OES.

The data of the ICP-OES and EDS analysis is shown in Figure R8 included in the supporting information as Figure S4.

We also determined the concentrations of the cations in the hydroxide electrolytes using ICP-OES. It turns out that the “dip” in RbOH is also reflected as a dip in the cation concentration (see Table R3).

However, it needs to be noted that there is a certain standard error attached with the ICP-OES data. First, the alkali metal cations are easily ionized elements (EIE) and suffers therefore from interference. Since the hydroxides were diluted quite significant, this effect is although minimized. We also ran control samples to better understand the no interference. Then, the cations all have different emission sensitivity; Li⁺ (~100,000 cts/ppm), Na⁺ (~10,000 cts/ppm), K⁺ (~5000 cts/ppm), Rb⁺ (~2000 cts/ppm). The cations also require different dilution factors since ICP-OES is measured in wt. % and not in at. %, and the molecular weight differs between the cations. Despite this, we measured very similar concentrations to the expected concentrations according to the pH-values. We also read a slightly lower than expected concentration of the Li⁺ cation in LiOH, however, Li⁺ should suffers most from the EIE effect, so it is unclear whether this is a real effect or not, and we would not put too much weight on this.

Table R3 is included in the Supplementary information as Table S7.

Table R3. (Table S7). ICP-OES of the alkali hydroxides to determine the concentration of the cations (Li⁺, Na⁺, K⁺, and Rb⁺). The concentration of for Cs⁺ could not be determined due to zero emission intensity. The electrolyte pH was calculated from the given concentration according to eqs. (S1)-(S13). It needs to be kept in mind that there is a certain standard error associated with these values, and these should rather be used as an approximation. This is because there is a significant difference in emission intensity and sensitivity between the alkali cations (Li⁺, Na⁺, K⁺, and Rb⁺) at comparable concentrations. Also, since the concentration is measured in ppm, (and not in molarity), it means that the different cations had to be diluted with a different factor. The differences in molecular weight further adds up to this error since the total dilution factor is relatively large (~5000x), and differs between the cations.

	Li	Na	K	Rb	Cs	Ni	Fe
c (M)	c cation from ICP-OES (M)						
0.1 (*purified)	0.061	0.094	0.085	0.079	n/a ^(a)	n/a ^(b)	n/a ^(b)
1 (*purified)	0.61	0.91	0.85	0.80	n/a ^(a)	n/a ^(b)	n/a ^(b)
	Calculated pH						
0.1 (*purified)	12.73	12.95	12.92	12.89	n/a ^(a)	n/a ^(b)	n/a ^(b)
1 (purified)	13.51	13.91	13.84	13.83	n/a ^(a)	n/a ^(b)	n/a ^(b)

^{n/a)} no value obtained or not analyzed, se label for more information.

^{a)} The Cs⁺ ion has zero emission intensity and cannot be determined by ICP-OES.

^{b)} The element could not be detected with ICP-OES, the values are below the detection limit.

We have also added this information to the Supplementary information and inserted a paragraph on p. 5 in the manuscript:

ICP-OES analysis of the as-received (non-purified) electrolytes indicated trace amounts of ~5 ppm Fe and ~3 ppm Zn in RbOH, and small impurities of Fe (≤ 1 ppm) in CsOH. These impurities were successfully removed in the purification step (see Figure S4a-b), and no other major impurities were found after purification, also confirmed by EDS analysis (see Figure S4c).

3. The iR correction needs to be explained better, and possibly redone. The authors state that measurements were done with “with 85 % iR-correction” and that “A typical value of the iR-drop in the in situ XAS setup was 25 Ohm”. I have two questions here: 1) What happened with the remaining 15% iR-loss? Was that left uncompensated? If so, that should have a marked effect on

the results of both pH and cation effect? 2) it appears strange to give one typical value for the uncompensated resistance when the alkali concentration changed almost two orders of magnitude, this ought to affect the uncompensated resistance a lot?

Indeed, the reviewer is right that the iR-drop affects the activity significant, and is an important parameter. We always carried out automatic iR-compensation independently to each catalyst film and every time we started a new measurement. The uncompensated solution resistance (R_u , iR-drop) was determined using electrochemical impedance spectroscopy (EIS) usually at 10 kHz where the phase approaches zero, and a level of 85 % iR-compensation was applied in the EC-Lab software of the Bio-Logic potentiostat (the "ZIR" method), both for the in situ XAS measurements and for the pH-series. We compensated only for a total level of 90 % (so additional 5 %) since otherwise clear overcompensation effects were often observed (see example in Figure R7 in this rebuttal letter). To meet the concerns of the reviewer, we have gone through all data again, to make sure we always applied a final level of 90 % total iR-compensation for all measurements. We did not see that this changed anything regarding the relative cation trends, however, small corrections to the absolute levels occurred in some figures.

We have clarified our approach regarding the iR-compensation in the Methods section, and we have moved this part of the Methods section to the main manuscript since it was appearing in the Supplementary Information before. It is named "**Correlations between the OER activity and pH investigated "in-house"**".

It would be nice to see a table in SI with the values of the uncompensated resistance in all measurements, especially the different alkali concentrations. 4. The authors should motivate and explain more on the pH measurements and how valid these are. From the text and Table S4 it is clear that no results were obtained for LiOH due to "Erroneous readings". Why was this happening? Could the authors not use another instrument or another technique? Additionally, in my experience, a "standard pH-meter" as the authors seem to use are often decently accurate around neutral pH, but for highly acidic or highly alkaline solutions the accuracy is often quite bad. Since pH is a crucial parameter in this work, I think more care and more precision should go into determining it.

We thank the reviewer for this comment, which has helped us to improve and strengthen the part that concerns the electrolyte pH of the hydroxides, as well as the validity of the pH-trends.

We have added the values of the uncompensated solution resistances (R_u , iR-drop) in the 0.1 M hydroxide solutions to Table S1 in the manuscript's supporting information and they are shown in this letter for reference (see Table R2), which contains mostly the OER activity parameters measured with the leak-free Ag/AgCl reference electrode used for the in situ XAS measurements. We have also included the values for the uncompensated resistance for the alkali hydroxides with different concentrations, namely 0.05 M to 1 M (see Table S8 in the manuscript's Supplementary information). The data is also provided in this rebuttal letter as a reference (see Table R4):

Table R4. (values are included in Table S8). Uncompensated solution resistance in the alkali hydroxides using the home-manufactured RHE reference electrode. The R_u was estimated using impedance spectroscopy at 10 kHz.

	LiOH	NaOH	KOH	RbOH	CsOH
c (M)			R_u (iR-drop)		
			(Ω)		
0.05	298 (6)	261 (12)	308 (6)	320 (10)	196 (8)
0.1	138 (5)	147 (8)	145 (2)	178 (6)	106 (3)
0.25	67 (1)	68 (4)	67 (3)	80 (3)	57 (1)
0.5	40 (1)	45 (5)	50 (7)	41 (2)	30 (1)
1	27 (1)	28 (1)	22 (1)	21 (1)	21 (1)

We ask the reviewer to notice that the iR-drops are different between the Ag/AgCl and the RHE reference electrodes. The iR-drop is higher for the RHE electrode, which partly depends on that the distance between the reference and the working electrode, but also because of the construction. For clarity and to validate the relative trends using either of the reference electrodes, we therefore thought it was necessary to make a more extensive evaluation and comparison between these two data sets (see Figure S1 in the manuscript's Supplementary information). This shows that the relative trends agree well independent of the reference electrode, despite that the absolute iR-drops are different. There are some differences in the absolute current densities between the two data sets, however, this does not seem to impact the cation trends.

We have changed the text in the manuscript on p. 3 to reveal these changes:

A larger data set recorded “in house” largely confirms the activity trend from the XAS data set, where the OER overpotential (η_{OER}) at 10 mA cm⁻² increases in the order of CsOH (295 mV) < NaOH (309 mV) < KOH (316 mV) < RbOH (331 mV) < LiOH (352 mV), see Figure 2a.⁶ Since the two data sets were measured using two different reference electrodes (Ag/AgCl vs. RHE), we further validate the trends in the Supplementary information Figure S1a-e.

Then, it is true that the “standard” pH-meter does not have as high accuracy in alkaline solutions as it does close to neutral pH-values. The absolute pH values will vary with the calibration curve, and will depend on which pH-electrode was used. Despite that the absolute pH values are not perfectly accurate; it seems as if the relative values are accurate even within the selected pH-range (pH 13-14). We confirmed this by checking the pH of commercial KOH solutions purchased from Merck, these have a standardized concentration (0.1 M KOH and 1 M KOH, Titripur, Reag. Ph. Eur). We measured pH 13.01 in 0.1 M KOH and 13.88 in 1 M KOH. The theoretical pH-values for these are 12.99 and 13.90, respectively, according to eqs. S(1)-S(13) provided in the Supplementary information. Therefore, the major issue seems to be a systematic error, which depends on the offset of the calibration curve. Since we only compare the relative pH-values in this study, this error should not affect the accuracy of the relative cation trend. We recently purchased a new pH-electrode that has a ceramic membrane and more suitable for the alkaline pH-range (InLab® Routine Pro-ISM from Mettler-Toledo). We did however not have the old hydroxide solutions saved, so we cannot further confirm the accuracy. However, according to some new control measurements of freshly prepared hydroxide solutions presented in Figure R2 in this rebuttal letter (and attached as new Figure S17 in the Supplementary information), the cation effect could be validated using this new pH-electrode as a likely pH-effect.

The next issue raised by the reviewer was the fact that we could not determine the pH of LiOH with the standard pH-meter. Even with the new specialized pH-electrode that is more suitable for higher

pH, we observed the same effect in LiOH as with the old pH-electrode. So, it is very challenging to obtain accurate pH-values when the Li^+ ion is present in solution. We therefore made efforts to find another method to obtain a pH-value of LiOH. What turned out as the most successful was simply to use the potential-difference between the reversible hydrogen electrode (i.e. pH-dependent) and an Ag/AgCl electrode (i.e. pH-independent), and constructed a calibration curve using commercial solutions with a known pH. The results are shown in as Figure R10, and has been added to the manuscript's Supplementary Information as Figure S16. The pH obtained in LiOH this way is indeed very close to the theoretical pH of LiOH (or it is slightly higher than expected at low concentrations). At higher concentrations, it is evident that the pH in LiOH increases much less compared to CsOH, which is expected from theory according to the $\text{p}K_b$ values and Eqs. S(1)-S(13), where the basicity increases from LiOH to CsOH.

Figure R10. (Figure S16). pH calibration curve using the potential difference between the reversible hydrogen electrode (RHE) and the leak-free Ag/AgCl reference electrode, obtained using commercial buffer solution pH 9 (Certipur[®], traceable to SRM from NIST and PTB, Merck), buffer solution pH 12 (Reagecon, traceable to NIST), and commercial KOH solution 0.1 M (pH 12.99) and 1 M (pH 13.90) (Titripur[®], Supelco). The pH of the commercial KOH solutions was assumed as the theoretical values according to eqs. (S1)-(S13). (b) The calculated pH in LiOH and CsOH using calibration curve presented in (a).

5. In the Results section it is stated that “A home manufactured reversible hydrogen electrode (RHE) was employed as reference electrode to avoid unknown potential-shifts due to the pH”. But in the Methods section it says that “a leak-free Ag/AgCl employed as reference electrode” and that the “offsets were determined by calibration against an RHE electrode”. This needs some clarification as it is quite a different thing to use an RHE electrode as reference electrode and to calibrate a Ag/AgCl electrode.

Thanks to the reviewer for pointing this out, we have edited the Methods section where we have now clarified which reference electrode was used for which measurements.

In short, for the in situ XAS measurements, a leak-free Ag/AgCl reference electrode was always used because it was simpler to have this electrode in the XAS setup at the beamline. For the

concentration/pH-series (0.05 M, 0.1 M, 0.25 M, 0.5 M, and 1 M), an RHE electrode was always used as reference. In fact, we used the RHE electrode for all “in house” measurements, including determining the OER overpotential at 10 mA cm⁻², potential steps, and Tafel slope collection where larger data sets were recorded during steady-state conditions. The offset of the leak-free Ag/AgCl reference electrode was determined by calibration against the RHE electrode (i.e. by determining the potential difference between the Ag/AgCl and the RHE electrode). The relative trend observed against both reference electrodes were similar across the cation series, and therefore we assumed that both electrodes were accurate to use. (The reason why we used an RHE electrode for the pH-series instead of the Ag/AgCl was because it was more convenient not having to determine the offset of the Ag/AgCl reference electrode in every new concentration and cation).

To put more focus on the validity between these two difference reference electrodes, we have now included a more extensive comparison in Figure S1 in the Supplementary information, where the in situ XAS measurements (Ag/AgCl) and the “in house” measurements (RHE) are compared. We have made sure to declare this in both the Methods section and the caption texts.

Minor Issues

- 1. First sentence in the introduction states that it is imperative to “slow-down the steadily growing carbon footprint”, it should be the growth that should be slowed down, the footprint does not really have a motion.***

Thanks to the reviewer for this suggesting, we have rephrased the sentence to following:

To slow-down the growth of the steadily increasing carbon footprint

Author's Response to Reviewer #4:

Currently, serious energy and environmental crises are the grand challenges facing society worldwide. The correlative investigations of electrocatalysts for water splitting process (e.g. OER and HER) have become one of the hottest areas. In this work, the authors investigate the OER activity of the Ni-Fe oxyhydroxide, where the activity increases in the order of $\text{LiOH} < \text{RbOH} < \text{KOH} \approx \text{NaOH} < \text{CsOH}$. They claim that the electrolyte pH (rather than the cation size) can play an important role in affecting the OER activity, and therefore the electrolyte pH can serve as an activity descriptor to determine the OER activity. This work is interesting, however, the following issues should be addressed:

1) From figs 7C and S16, it can be found that many hydrogen bonds exist in these studied structures. Therefore, a semi-empirical van der Waals (vdW) correction must be employed to account for the correlative dispersion interactions for this kind of systems. However, it cannot be judged whether the vdW correction has been added during the computational process. Thus, authors must clarify it in view of the reliability of computational results.

We thank the reviewer for bringing this point up, and in fact we totally agree. Dispersion plays a role in these structures, although the validity of including semi-empirical dispersion corrections is a matter of continuous debate within the computational society. We have tested the effect of including D3(BJ) dispersion. This leads to contracted structures compared to the case of no explicit dispersion (see Table S12 in the manuscript's Supporting Information). However, the local reactivity properties are only moderately influenced, it is shown in Table S11 in the supporting information of the manuscript. We judge that dispersion effects are negligible for the computational approach we are using in this study.

Based on the above, we include the following paragraph in the main article on page 15, at the end of the computational details section:

The robustness of the computed results was tested by evaluating the effect of including empirical dispersion corrections via the D3(BJ) method,^{17,18} as well as implicit solvation following the VASPSOL¹⁹ protocol. The effect on the predicted reactivities is negligible for dispersion corrections on the predicted reactivities is negligible (see Table S11-12). The solvation alters the absolute values of the local reactivity properties, whereas the trends (and thus the conclusion) over the series of cations is the same with or without solvation (see Table S13-S14).

2) The authors should consider the effect of solvation on the correlative computed results, in view of the fact that the OER reaction occurs in water as a solvent

Thanks to the reviewer for this suggestion. Unfortunately, our analysis of local reactivity properties cannot be combined with a full explicit description of solvation. We evaluate the properties on isodensity surfaces that become masked by explicit water molecules. However, the approach can be used together with implicit solvation methods. We therefore implemented our methods in conjugation with the VASPSOL (J. Chem. Phys. 140, 084106, 2014) solvation method. We find that the absolute values of the local properties change, for some sites significantly (Table S13). This is well in line with previous studies of solvation effects on molecular systems (e.g., Adv. Theory Simul. 2019, 2, 1800149). Nonetheless, the trends (and thus the conclusion) over the series of cations remains the same regardless if solvation is used or not. This renders us confident that our previous analysis of the intrinsic reactivity of the system is accurate.

A comment about this is included in the manuscript on page 15 (see our reply to point 1 raised by this reviewer).

3) For the convenient understanding, the authors should clarify the definition of “Fe edge site” in detail.

We thank the reviewer for this suggestion. We discovered a small misunderstanding regarding which sites was actually modelled in Site 1 and Site 2. Both Site 1 and Site 2 are defined as “edge sites”, and are always surrounded by 2 μ_2 -O bridged oxygens and 3 μ_3 -O bridged oxygens, i.e. they are coordinatively unsaturated. There is therefore no real significant difference between Site 1 and Site 2 with respect to the structural geometry. Note that due to the positions of the alkali cations and the local water coordination, there is a significant difference between site 1 and 2, which is why we have decided to analyze them separately. All other “lattice sites” are coordinatively saturated, and are surrounded by 6 μ_3 -O bridged oxygens. We demonstrate these sites in Figure R11 in this letter, and they are also included a in the manuscript’s Supplementary information Figure S21. We have made sure to change the text in the manuscript accordingly, since we cannot draw conclusions regarding the difference in the reactivity between “edge” and “lattice” sites. Otherwise, this did not change anything regarding the relative cation trends, and more focus is not put on that.

Figure R11. (Figure S21.) Surface model of Fe-doped Ni(Fe)OOH(100) (b) The coordination geometry for Site 1 and Site 2 in the model. The main difference between site 1 and 2 is the coordination environment of H₂O and the alkali metal cation. Both Site 1 and Site 2 are “edge sites” surrounded by 2 μ_2 -O and 3 μ_3 -O bridged oxygen sites, and all the other lattice sites in the model have 6 coordinatively saturated μ_3 -O sites.

REVIEWERS' COMMENTS

Reviewer #1 (Remarks to the Author):

The authors have done a good job revising the manuscript. All my concerns have been addressed. It is recommended to be published in Nature Communications.

Reviewer #2 (Remarks to the Author):

The authors have done a thorough job to answer the raised comments and have significantly improved the manuscript. I have no further questions and I suggest the manuscript to be published.